# The human thalamus orchestrates neocortical oscillations during NREM sleep

Thomas Schreiner ®[1], Elisabeth Kaufmann[2], Soheyl Noachtar[2], Jan-Hinnerk Mehrkens[3] & Tobias Staudigl ®[1] ✉

A hallmark of non-rapid eye movement sleep is the coordinated interplay of slow oscillations (SOs) and sleep spindles. Traditionally, a cortico-thalamo-cortical loop is suggested to coordinate these rhythms: neocortically-generated SOs trigger spindles in the thalamus that are projected back to neocortex. Here, we used intrathalamic recordings from human epilepsy patients to test this canonical interplay. We show that SOs in the anterior thalamus precede neocortical SOs (peak −50 ms), whereas concurrently-recorded SOs in the mediodorsal thalamus are led by neocortical SOs (peak +50 ms). Sleep spindles, detected in both thalamic nuclei, preceded their neocortical counterparts (peak −100 ms) and were initiated during early phases of thalamic SOs. Our findings indicate an active role of the anterior thalamus in organizing sleep rhythms in the neocortex and highlight the functional diversity of thalamic nuclei in humans. The thalamic coordination of sleep oscillations could have broad implications for the mechanisms underlying memory consolidation.

The presence and coordinated interplay of slow oscillations (SOs) and sleep spindles hallmarks non-rapid eye movement (NREM) sleep[1]. SOs (~1 Hz) reflect neuronal network alterations of the membrane potential between periods of neuronal silence ('hyperpolarization', i.e., down-state) and neuronal excitation (depolarization, i.e., up-state)[2,3]. The origin of SOs has been traditionally located to neocortical circuits[3,4], triggering time windows of excitability and inhibition not only in neocortex but also in the thalamus and the hippocampus[3]. However, evidence from animal models casts doubt on the view that SOs are exclusively initiated and orchestrated by neocortical activity and suggest that the thalamus plays a critical role in synchronizing and coordinating SO activity[5–10].

The depolarizing phase of SOs is assumed to initiate the generation of sleep spindles within the thalamic circuitry[11]: reciprocal interactions between the thalamic reticular nucleus and thalamocortical neurons result in waxing and waning oscillations in the range of 11–16 Hz. Sleep spindles are usually nested towards the excitable up-states of neocortical SOs[12–14], and are also found in the hippocampus, where they are thought to synchronize hippocampal ripples[15,16].

Importantly, this triple-coupling of sleep-related oscillations has been suggested to facilitate memory consolidation, by synchronizing neuronal activity across brain regions and relaying memory representations between the hippocampus and neocortical long-term stores[17]. While a plethora of studies bolster the critical role of sleep oscillations and their coordinated interplay for the memory function of sleep[1,18], the exact neural circuits enabling their complex orchestration across brain areas are less clear, ultimately impeding our understanding of the prime mechanistic vehicle of memory consolidation.

This is particularly striking with respect to the neural circuits facilitating memory consolidation in humans, since direct neural recordings from one of the key players, the thalamus, are scarce. The anterior thalamic nuclei (ANT) and the mediodorsal thalamus (MD) have recently taken center stage as thalamic key areas for different

[1]Department of Psychology, Ludwig-Maximilians-Universität München, Munich, Germany. [2]Epilepsy Center, Department of Neurology, Ludwig-Maximilians-Universität München, Munich, Germany. [3]Department of Neurosurgery, Ludwig-Maximilians-Universität München, Munich, Germany. ✉e-mail: tobias.staudigl@lmu.de

aspects of human memory functions[19–21], and might therefore also play a key role in coordinating sleep oscillations relevant for memory consolidation. Leveraging the rare opportunity to record intracranial electroencephalography (iEEG) from the human ANT and MD, together with simultaneous scalp electroencephalography (EEG), we investigated the interplay of NREM sleep oscillations within a thalamocortical network that is known to serve memory functions during wake[22].

We show that SOs in the ANT, but not the MD, lead neocortical SOs. These results undermine the notion of an exclusive generation of SOs in the human neocortex and highlight the functional diversity of specific thalamic nuclei. Sleep spindles in both the ANT and MD preceded neocortical spindles, in line with their thalamic origin[23]. Furthermore, we show that the nesting of sleep oscillations in the thalamus differs from the nesting in neocortex. Thalamic spindles locked to earlier thalamic SO-phases than their neocortical counterparts.

## Results

To examine thalamocortical interactions during human NREM sleep, we analyzed simultaneously recorded intracranial thalamic and scalp EEG collected across 11 full nights of sleep from 8 patients with pharmaco-resistant epilepsy. On average, participants slept for $8.82 \pm 1.41$ h, with $63.51 \pm 0.07\%$ spent in NREM sleep (stages N2 and N3; see Supplementary Table 1 for proportions of times spent in each stage; Fig. 1c). Thalamic activity was recorded from the ANT and the MD (see Fig. 1a and methods for details on electrode localization). We isolated sleep oscillations from 27 bipolar intracranial thalamic channels (12 ANT; 15 MD) and scalp EEG electrodes (see Supplementary Table 2 for details). To determine thalamocortical coordination during NREM sleep and their specificity, we detected SOs, spindles and SO-spindle events independently in thalamic iEEG (ANT & MD) and scalp EEG recordings, assessed their coupling and delineated the temporal relationship of thalamocortical interactions (for descriptive values and characteristics see Supplementary Tables 3–6; Fig. 1b).

## ANT SOs precede neocortical SOs

ANT SOs significantly preceded neocortical SOs, as revealed by occurrence probabilities of ANT SOs relative to neocortical SOs (Fig. 2a, positive cluster from −0.15 to 0.05 s, $p < 0.005$; tested against event-free occurrence probabilities; corrected for multiple comparisons across time; neocortical SOs were derived from frontal scalp EEG electrodes, see Supplementary Table 2 for details). ANT SO occurrence preferentially peaked, on average, briefly before the emergence of neocortical SOs (time of peak: −0.05 s). To further validate this outcome, we specifically determined the phase of ANT SOs for all paired SO-events (i.e., all ANT SOs within ± 750 ms of neocortical SOs) at the time of neocortical SO down-states (i.e., thalamocortical SO phase–phase coupling). We found significant nonuniform distributions in each of the ANT contacts (12/12) ($p < 0.05$, corrected for multiple comparisons using the False Discovery Rate (FDR)[24], Rayleigh test, mean vector length: $0.46 \pm 0.05$). Moreover, we found a significant nonuniform distribution across contacts (Rayleigh $z = 10.95$, $p < 0.0001$), with the phase of ANT SOs being ahead of their neocortical counterparts (the phase of neocortical SO down states corresponds to ± π; mean coupling direction: $-176.39 \pm 4.94°$; see Fig. 2b).

To showcase the generic features of ANT and neocortical detected SOs, time–frequency representations (TFRs) of SO-locked neocortical (Fig. 2c) and ANT (Fig. 2e) activity were contrasted against event-free events. TFRs time locked to scalp-derived SO down-state peaks (see Supplementary Table 2 for details) exhibited the prototypical modulation of low frequency (<5 Hz) and spindle power (11–16 Hz) during SO events (Fig. 2c). In particular, low frequency power peaked before the SO down-state. In contrast, spindle power was diminished during the down-state, increased during the positive deflections of the SO and peaked during the SO up-state (significant differences: $p < 0.05$, corrected for multiple comparisons across time and frequency). Remarkably, while low frequency power for ANT-derived SOs showed similar pre-down-state increases, power in the spindle band (11–16 Hz) and beyond (>20 Hz) showed an early power increase tightly locked to the ANT SO down-state (Fig. 2e; $p < 0.05$, corrected for multiple comparisons across time and frequency). This

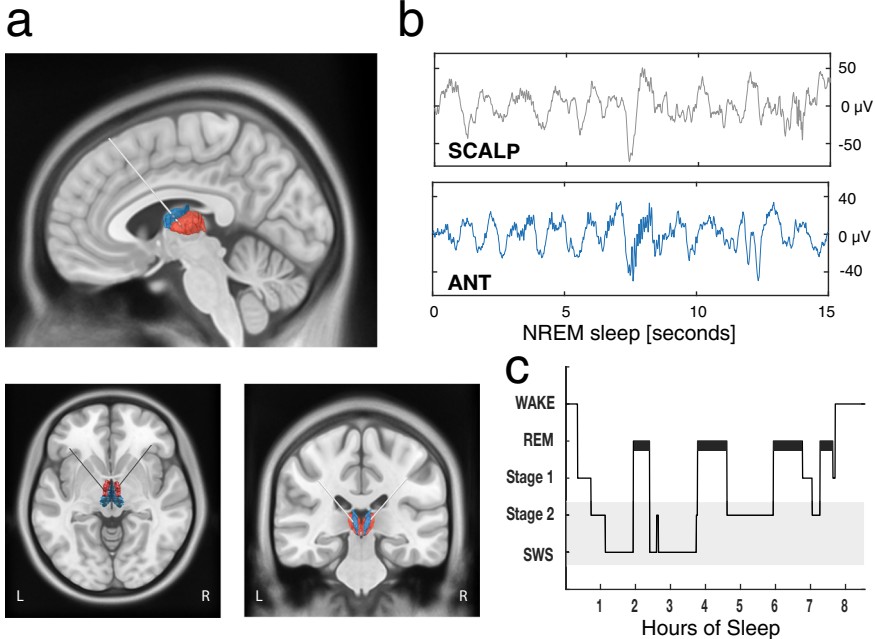

**Fig. 1 | Thalamic electrode placement and sleep architecture. a** Electrodes were implanted in the left and right ANT (blue) and MD (red). Localization of an exemplar patient's DBS leads (ref. 71) is shown on a T1-weighted template MRI with superimposed thalamus atlas (ref. 74). **b** Example of NREM sleep segment (15 s), comprising SOs and sleep spindles (top row: scalp recording; bottom row: ANT). **c** Hypnogram of a sample participant, showing time spent in different sleep stages across one recording night. The gray shading indicates NREM sleep stages N2 and SWS.

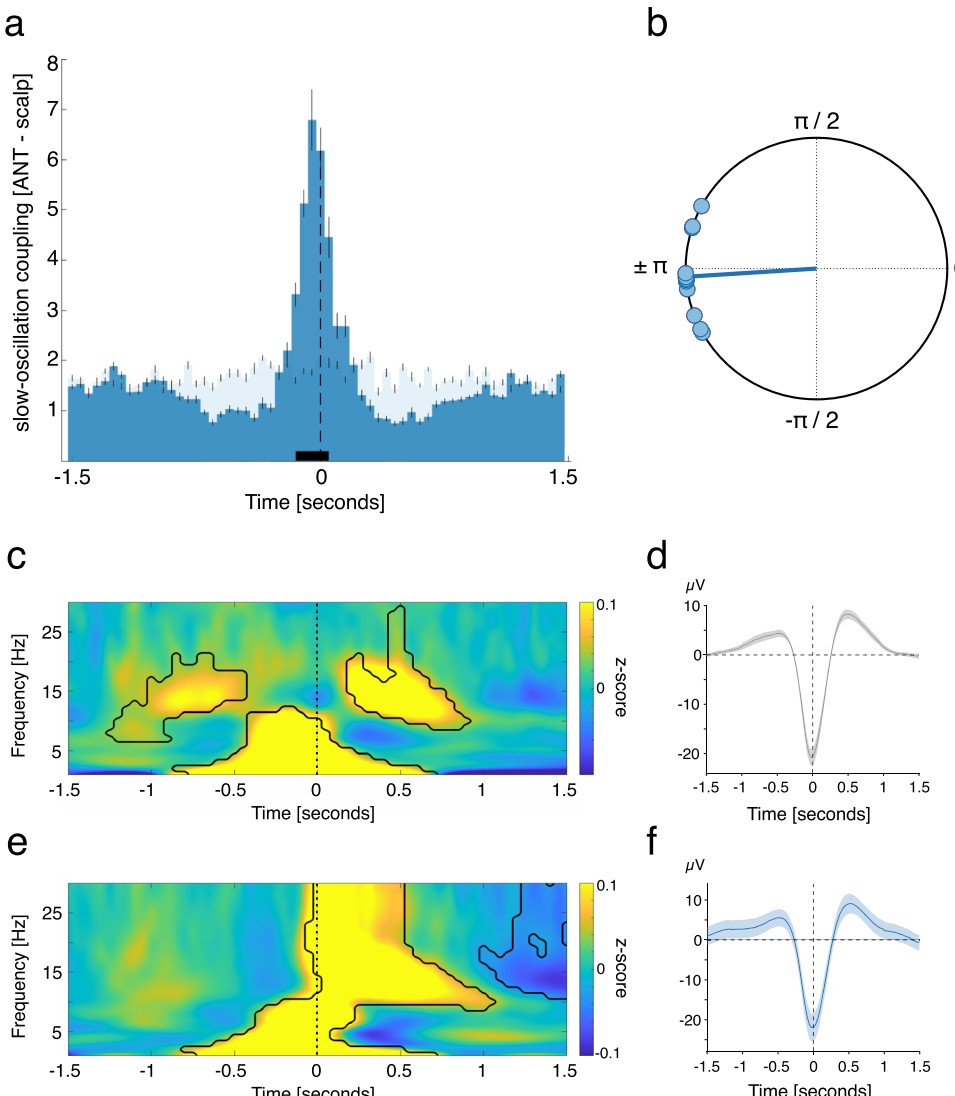

**Fig. 2 | ANT SOs precede neocortical SOs. a** Occurrence probabilities of ANT SO down-state peaks relative to neocortical SO down-state peaks (time = 0; dashed line; bin size = 50 ms), indicating that ANT SOs precede neocortical SOs. The solid black bar indicates significant differences, resulting from comparison with SO-free control events as derived from a dependent-samples $t$-test (two-sided, positive cluster from −0.15 to 0.05 s, $p < 0.005$; time of peak: −0.05 s, cluster corrected across time). **b** Phase of ANT SOs at the time of neocortical SO down-states for paired SO–SO events, illustrating that ANT SO phases preceded their neocortical counterparts (phase of neocortical SO down-states corresponds to ± π; mean coupling direction: −176.39 ± 4.94°; Rayleigh test, one-sided: $p < 0.0001$; $z = 10.95$). **c** Time–frequency representation of neocortical SOs (locked to neocortical SO down-states), contrasted against event-free segments. The contour lines indicate clusters (two-sided dependent-samples $t$-test; $p < 0.05$, corrected for multiple comparisons across time and frequency). **d** Grand average EEG trace of neocortical SOs (mean ± SEM, negative peak, time 0, $N = 1573.8 ± 129.9$). **e** Time–frequency representation of all ANT SOs (locked to ANT SO down-states), contrasted against event-free segments. The contour lines indicate clusters ($p < 0.05$, corrected for multiple comparisons across time and frequency). **f** Grand average iEEG trace of ANT SOs (mean ± SEM, negative peak, time 0, $N = 1513.5 ± 130.1$). Source data are provided as a Source Data file.

pattern deviates from generic modulations of spindles by SOs as described in our and previous scalp EEG recordings, where spindles nest towards SO up-states[14]. Figure 2d illustrates the scalp EEG grand average, locked to the minimum of neocortical SO down-states. Figure 2f illustrates the ANT grand average locked to the minimum of ANT SO down-states.

**Neocortical SOs precede SOs in the MD**
Next, we asked, whether SOs leading neocortical SOs is specific to the ANT or can also be found in other thalamic nuclei. To address this question, we investigated the interplay between neocortical SOs and SOs in the mediodorsal thalamus (MD).

In stark contrast to ANT SOs, SOs in the MD did not precede, but were on average led by neocortical SOs, as evidenced by occurrence probabilities (Fig. 3a; positive cluster from −0.05 to 0.1 s, $p = 0.021$;

corrected for multiple comparisons across time; time of peak: 0.05 s; neocortical SOs were derived from frontal scalp EEG electrodes, see Supplementary Table 2 for details). Again, we determined the phase of thalamic SOs for all paired SO-events at the time of neocortical SO down-states (± 750 ms). We found significant nonuniform distributions in all MD contacts (15/15, $p < 0.05$, corrected for multiple comparisons using FDR; Rayleigh test, mean vector length: 0.41 ± 0.06) and a significant nonuniform distribution across contacts (Rayleigh $z = 7.94$, $p = 0.0001$), with the phase of MD SOs following their neocortical counterparts (mean coupling direction: 133.51 ± 10.91°; see Fig. 3b). TFRs time-locked to scalp-derived SO down-states exhibited the typical modulation of low frequency (<5 Hz) and spindle power (11–16 Hz) during SO events (Fig. 3c; $p < 0.05$, corrected for multiple comparisons across time and frequency). Similar to the ANT, TFRs locked to the down-states of MD SOs exhibited significant low-frequency power

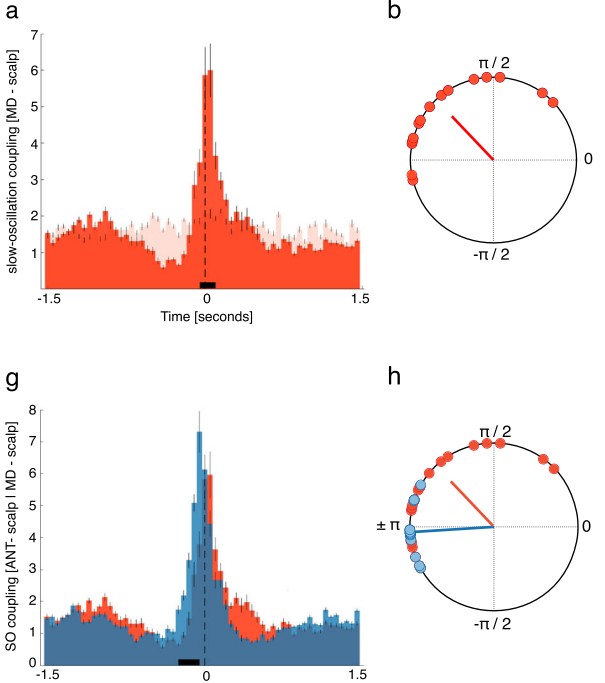

Fig. 3 | **Neocortical SOs precede MD SOs. a** Occurrence probabilities of MD relative to neocortical SO down-states (time = 0, dashed line; bin size = 50 ms). On average, neocortical SOs lead MD SOs. The solid black line indicates significant differences, compared to SO-free events (dependent-samples $t$-test, two-sided, cluster: −0.05 to 0.1 s, $p = 0.021$; peak time: 0.05 s, corrected across time). **b** Phase of MD SOs during neocortical down-states for paired events. MD SO phases followed neocortical counterparts (± π indicates phase of neocortical down-states; mean coupling direction: 133.51 ± 10.91°; Rayleigh test, one-sided: $p < 0.0001$; $z = 7.94$). **c** Time–frequency representation (TFR) of neocortical SOs (locked to down-states), contrasted against event-free segments. Contour lines indicate clusters (two-sided dependent-samples $t$-test, $p < 0.05$, corrected across time and frequency). **d** Grand average of neocortical SOs (mean ± SEM, negative peak, time 0, $N = 1543.4 \pm 114.9$). **e** TFR of all MD SOs (locked to down-states), contrasted against event-free segments. Contour lines indicate clusters (two-sided dependent-

samples $t$-test, $p < 0.05$, corrected across time and frequency). **f** Grand average of MD SOs (mean ± SEM, $N = 1453.9 \pm 121.3$). **g** Occurrence probabilities for ANT-neocortical (blue) and MD-neocortical SO interactions (red). ANT SOs emerged before MD SOs with regards to their neocortical counterparts (dependent-samples $t$-test, two-sided, cluster: −0.25 to −0.05 s, $p = 0.018$; corrected across time). **h** Significant difference in phase distribution for ANT vs. MD related SO coupling (Watson–Williams test, two-sided: $F = 12.7$, $p = 0.0015$). **i** Occurrence probabilities of ANT relative to MD down-states. ANT SOs emerged before MD SOs (dependent-samples $t$-test, two-sided, positive cluster from −0.15 to 0 s, $p = 0.015$; peak time: −0.05 s, corrected across time). **j** Phase of ANT SOs at the time of MD SO down-states for paired SO–SO events. ANT SO phases preceded their MD counterparts (±π indicates phase of MD down-states; mean coupling direction: −138.34 ± 11.08°; Rayleigh test, one-sided: $p < 0.0001$; $z = 7.44$). Source data are provided as a Source Data file.

increases before the down-state peak (Fig. 3e, time = 0), and power increases in the sleep spindle band and beyond locked to the down-state (significant difference: $p < 0.05$, corrected for multiple comparisons across time and frequency). Figure 3d illustrates the scalp EEG grand average, locked to the minimum of neocortical SO down-states. Figure 3f illustrates the MD grand average locked to the minimum of MD SO down-states.

Next, we compared ANT-neocortical and MD-neocortical interactions with regards to SOs and tested whether ANT SOs would systematically emerge earlier (in relation to neocortical SOs) than MD SOs. The occurrence probabilities of ANT SOs relative to neocortical SOs differed significantly from those obtained from MD SOs relative to neocortical SOs, with ANT SOs preferentially emerging before MD SOs (Fig. 3g; positive cluster from −0.25 to −0.05 s, $p = 0.018$; corrected for multiple comparisons across time). Also, the phase distribution for the preferential coupling between thalamus and neocortex differed significantly when comparing ANT and MD related SO coupling (Fig. 3h; Watson–Williams test: $F = 12.7$, $p = 0.0015$), supporting the leading role of ANT SOs as compared to the MD. Finally, we directly assessed ANT − MD related SO interactions and tested whether ANT SOs would precede MD SOs (here occurrence probabilities were obtained for all possible ANT–MD combinations per participant, leading to 13 distributions). In line with the previous findings, results of this direct comparison revealed that ANT SOs preferentially emerged before MD SOs (positive cluster from −0.15 to 0 s, $p = 0.015$; time of peak: −0.05 s;

Fig. 3i). Again, we obtained the phase of ANT SOs for all paired SO-events at the time of MD SO down-states (±750 ms). We found a significant nonuniform distribution across contacts (Rayleigh $z = 7.44$, $p = 0.0001$), with the phase of ANT SOs preceding their MD counterparts (mean coupling direction: −138.34 ± 11.08°; see Fig. 3j).

## ANT spindles precede neocortical spindles

Animal models established that sleep spindles are generated in the thalamus and spread to the neocortex along thalamocortical fibers[23]. However, it is unclear which of the many thalamic nuclei contribute to the generation of spindles and whether this translates to humans. Here, we show that the human ANT is putatively part of the spindle generating thalamic circuit, as ANT spindle occurrence peaked on average before the emergence of neocortical spindles (Fig. 4a; positive clusters: −0.25 to 0.05 s, $p < 0.005$ and 0.15 to 0.2 s, $p = 0.019$; time of peak: −0.1 s; corrected for multiple comparisons across time; neocortical spindles were derived from frontal, central or parietal scalp EEG electrodes, see Supplementary Table 2 for details).

We further assessed the interplay of ANT and neocortical sleep spindles using a complementary analytical approach. Specifically, we tested to what extend spindle-related oscillatory power modulations would precede those in the neocortex, by computing power–power correlations in the sleep spindle band (see methods for details). We found a significant, off-diagonal cluster of temporal cross-correlations with regards to spindle power between ANT and neocortical sites

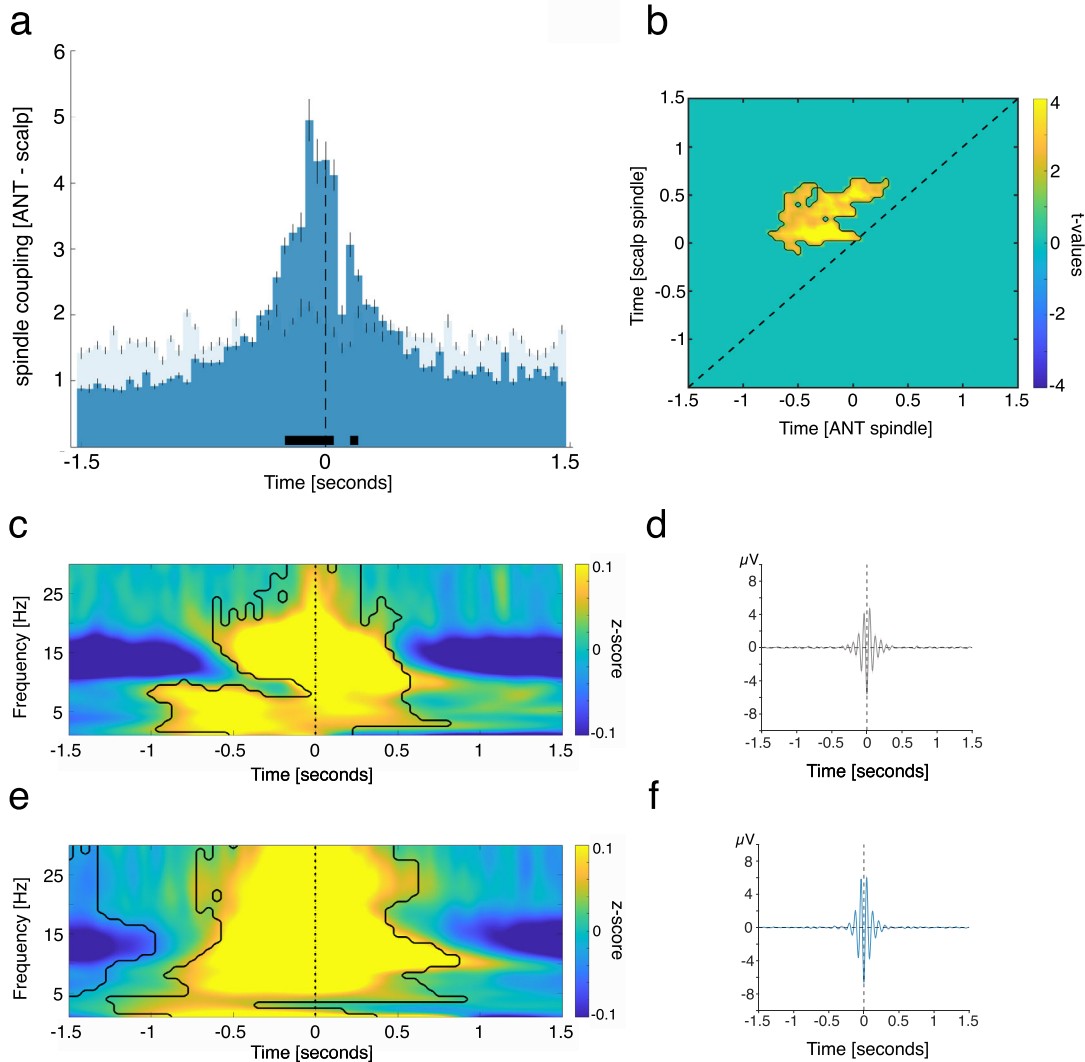

**Fig. 4 | ANT spindles lead neocortical spindles. a** Occurrence probabilities of ANT spindle peaks relative to neocortical spindle peaks (maximal negative amplitude, time = 0; dashed line; bin size = 50 ms), indicating that ANT spindles precede neocortical spindles. The solid black line indicates significant differences, resulting from comparison with spindle-free control events (dependent-samples *t*-test, two-sided, positive clusters from −0.25 to −0.05 s, *p* < 0.005 & 0.15 to 0.2 s, *p* = 0.019; time of peak: −0.1 s; corrected for multiple comparisons across time).
**b** Power–power correlations in the sleep spindle band for paired spindle events (locked to neocortical spindle peaks), contrasted against event-free correlation maps. The contour lines indicate the cluster (dependent-samples *t*-test, two-sided, *p* = 0.004, cluster corrected across time), depicting that ANT spindle power precedes neocortical spindle power. **c** Time–frequency representation of neocortical spindles (locked to neocortical spindle peaks), contrasted against event-free segments. The contour lines indicate clusters (dependent-samples *t*-test, two-sided, *p* < 0.05, corrected for multiple comparisons across time and frequency). **d** Grand average EEG trace of neocortical spindles (mean ± SEM, maximally negative amplitude, time 0, *N* = 1903.3 ± 211.5). **e** Time–frequency representation of all ANT spindles (locked to ANT spindle peaks), contrasted against event-free segments. The contour lines indicate clusters (dependent-samples *t*-test, two-sided, *p* < 0.05, corrected for multiple comparisons across time and frequency). **f** Grand average iEEG trace of ANT spindles (mean ± SEM, maximally negative amplitude, time 0, *N* = 2203.8 ± 213.5). Source data are provided as a Source Data file.

(*p* = 0.004, cluster corrected across time; Fig. 4b), supporting our initial finding that ANT spindles lead their neocortical counterparts. Figure 4c, e depict the spindle-locked (time 0 = maximally negative amplitude) neocortical (Fig. 4c) and ANT (Fig. 4e) related TFRs. While the power of scalp derived spindles was confined to the classical sleep spindle range (~11–16 Hz; Fig. 4c), spindle-related power increases in case of ANT recordings were also evident in higher frequencies (>20 Hz; Fig. 4e; *p* < 0.05, corrected for multiple comparisons across time and frequency). Figure 4d, f illustrate the grand average electro-physiological traces across scalp electrodes/ANT contacts (mean ± SEM, respectively), locked to the neocortical (Fig. 4d) and ANT (Fig. 4f) sleep spindle peaks (maximally negative amplitude, time 0). Figure 4d illustrates the scalp EEG grand average, locked to the neocortical sleep spindle peak. Figure 4f illustrates the ANT grand average, locked to the ANT sleep spindle peak. See Supplementary Fig. 1 for results

concerning spindles detected simultaneously in ANT and MD and spindles detected in ANT only.

## MD spindles lead neocortical spindles
Next, we tested the temporal relationship between MD and neocortical spindles. MD spindles occurred on average before neocortical spindles (Fig. 5a; positive clusters: −0.3 to 0.05 s, *p* < 0.005; time of peak: −0.05 s; corrected for multiple comparisons across time; neocortical spindles were derived from central or parietal scalp EEG electrodes, see Supplementary Table 2 for details). We also assessed power–power correlations in the sleep spindle band for neocortical- MD paired spindle events (locked to neocortical spindle peaks). We found a significant, off-diagonal cluster of temporal cross-correlation with regards to spindle power between MD and neocortical sites (Fig. 5b; tested against event-free segments; *p* = 0.002, cluster corrected across

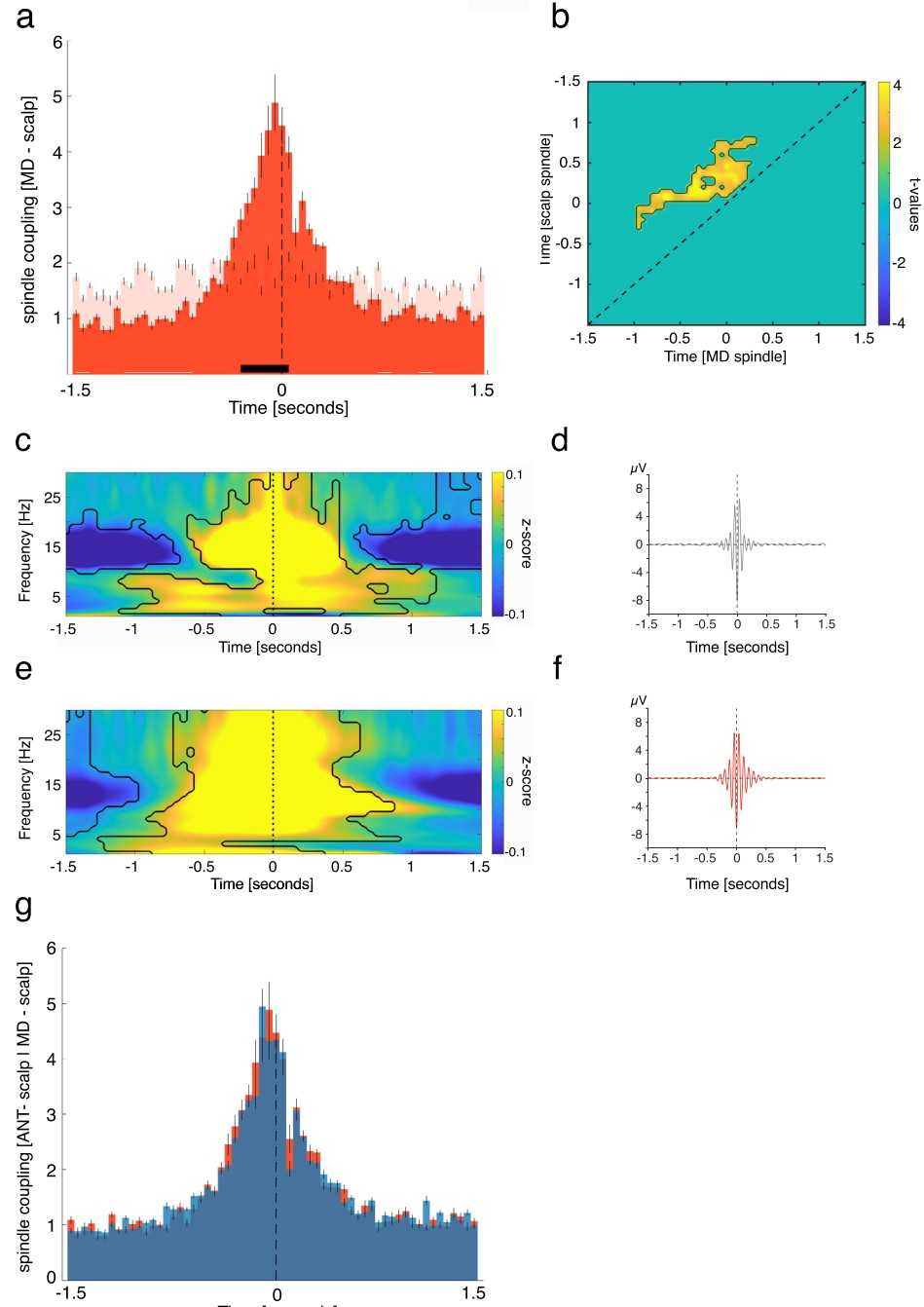

**Fig. 5 | MD spindles lead neocortical spindles. a** Occurrence probabilities of MD spindle peaks relative to neocortical spindle peaks (maximal negative amplitude, time = 0; dashed line; bin size = 50 ms), indicating that MD spindles precede neocortical spindles. The solid black line indicates significant differences, resulting from comparison with spindle-free control events (dependent-samples *t*-test, two-sided, positive cluster from −0.3 to 0.05 s, *p* < 0.005; time of peak: −0.05 s; corrected for multiple comparisons across time). **b** Power–power correlations in the sleep spindle band for paired spindle events (locked to neocortical spindle peaks), contrasted against event-free correlation maps. The contour lines indicate the cluster (dependent-samples *t*-test, two-sided, *p* = 0.002, cluster corrected across time), showing that MD-related spindle power precedes neocortical spindle power. **c** Time−frequency representation of neocortical spindles (locked to neocortical spindle peaks), contrasted against event-free segments. The contour lines

indicate clusters (dependent-samples *t*-test, two-sided, *p* < 0.05, corrected for multiple comparisons across time and frequency). **d** Grand average EEG trace of neocortical spindles (mean ± SEM, maximally negative amplitude, time 0, *N* = 1464.5 ± 252.6). **e** Time−frequency representation of all MD spindles (locked to MD spindle peaks), contrasted against event-free segments. The contour lines indicate clusters (dependent-samples *t*-test, two-sided, *p* < 0.05, corrected for multiple comparisons across time and frequency). **f** Grand average iEEG trace of MD spindles (mean ± SEM, maximally negative amplitude, time 0, *N* = 2093.1 ± 229.3). **g** Direct comparison of ANT-neocortical (blue) and MD-neocortical interactions (red) with regards to spindles yielded that the timing of ANT and MD-related spindles in relation to neocortical spindles was highly similar (dependent-samples *t*-test, two-sided, cluster with lowest *p* = 0.52). Source data are provided as a Source Data file.

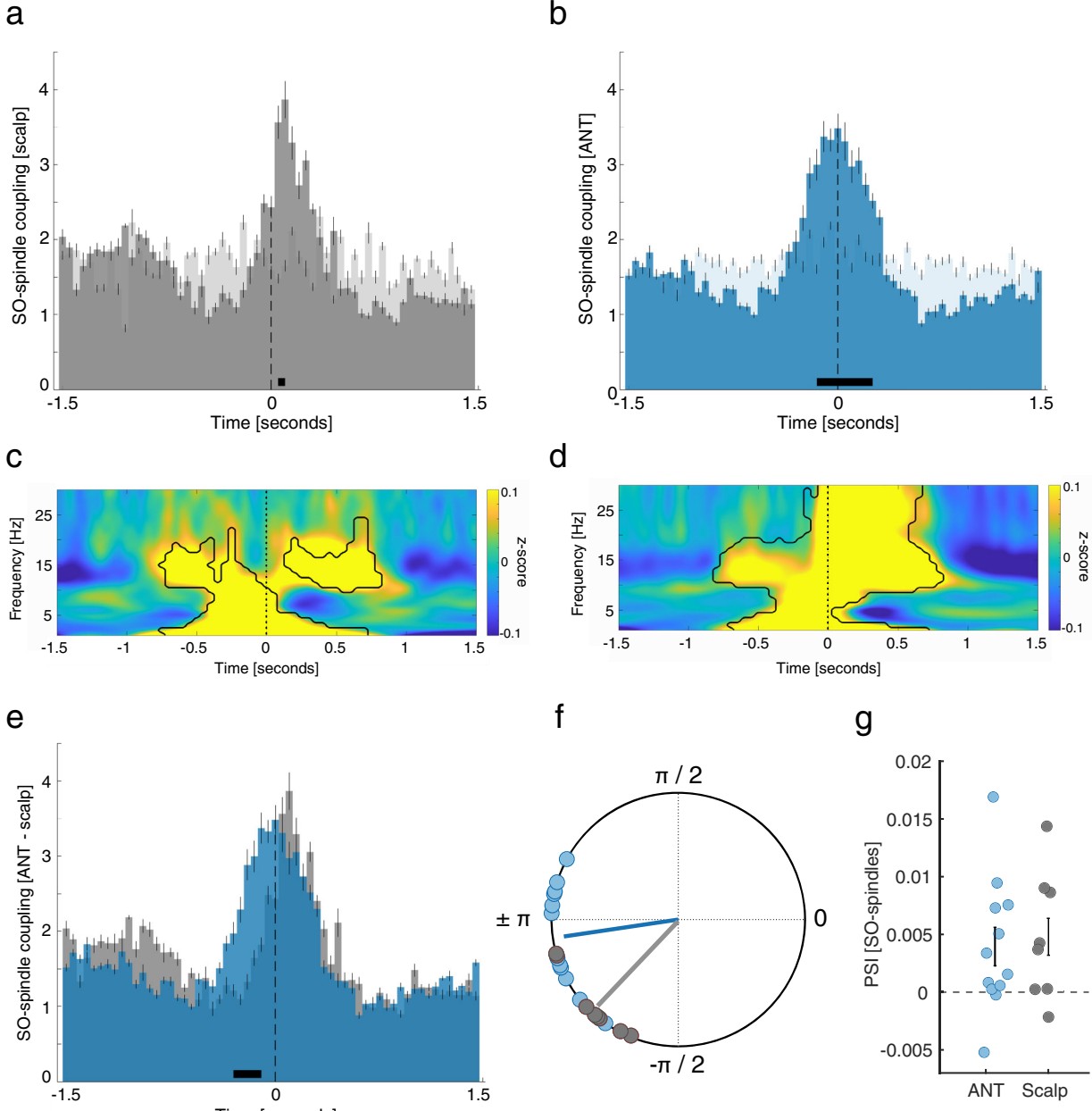

**Fig. 6 | ANT spindles lock to early SO phases. a** Occurrence probabilities of neocortical spindle onsets with respect to neocortical SO down-states (bin size = 50 ms), illustrating that spindles started at the SO down-to-up transition (0.05–0.15 s; dependent-samples *t*-test, two-sided, *p* < 0.0001; peak: 0.1 s; cluster corrected across time). **b** ANT spindles emerged around ANT SO down states (−0.15 to 0.25 s; dependent-samples *t*-test, two-sided, *p* = 0.0012; peak: 0 s; cluster corrected across time). **c** Time-frequency representation (TFR) of neocortical SO-spindle events (*N* = 304.1 ± 69.8; locked to neocortical SO down-states), contrasted against event-free segments. The contour lines indicate clusters (dependent-samples *t*-test, two-sided, *p* < 0.05, cluster corrected across time and frequency). **d** TFR of all ANT SO-spindle events (*N* = 431 ± 52.1, locked to ANT SO down-states), contrasted against event-free segments. The contour lines indicate clusters (dependent-samples *t*-test, two-sided, *p* < 0.05, cluster corrected across time and frequency). **e** Comparing the occurrence probabilities of ANT (blue) and neocortical SO-spindle complexes (gray) yielded that ANT spindles locked significantly

earlier to SOs as compared to neocortical spindles (−0.3 to −0.1, dependent-samples t-test, two-sided, *p* = 0.0039; cluster corrected across time). **f** Phases of the SO-spindle modulation derived from neocortical (gray) and ANT (blue) events. Neocortical spindle onsets emerged at the neocortical SO down-to-up transition (neocortical SO down-state phase corresponds to ± π; mean coupling direction: −137.14 ± 7.58°; Rayleigh test, one-sided: *z* = 6.91, *p* < 0.0001), while ANT spindles started around the ANT SO down-states (mean coupling direction: −175.22° ± 6.92°; Rayleigh test, one-sided: *z* = 9.33, *p* < 0.0001). The preferred phases of SO-spindle modulation differed significantly between the ANT and neocortical sites (Watson–Williams test, two-sided: *F* = 9.31; *p* = 0.0069). **g** Directional SO-spindle coupling as obtained by the phase-slope index (PSI; mean ± SEM) for ANT (blue) and neocortical (gray) SO-spindles. SO-phases significantly predicted spindle amplitudes both in case of ANT (*t*-test against zero, two-sided: *p* = 0.037) and neocortical SO-spindle events (*t*-test against zero, two-sided: *p* = 0.044).

time), indicating that MD spindle-related power modulations preceded neocortical spindle-related power.

Figure 5c, e depict the spindle-locked (maximally negative amplitude) neocortical and MD-related TFRs. Power of scalp-derived

spindles was, as expected, confined to the classical sleep spindle range (~11–16 Hz; Fig. 5c), while TFRs related to MD spindles exhibited power increases in the spindle band and beyond (>20 Hz). Overall, the relationship between MD spindles and neocortical spindles was highly

similar to the relationship between ANT spindles and neocortical spindles (see Fig. 4a, b). Figure 5d illustrates the scalp EEG grand average, locked to the neocortical sleep spindle peak. Figure 5f illustrates the MD grand average, locked to the MD sleep spindle peak. Figure 5g shows a comparison of ANT-neocortical and MD-neocortical interactions with regards to spindles. The timing of ANT and MD-related spindles in relation to neocortical spindles was highly similar. No significant differences emerged (cluster with lowest $p = 0.52$). See Supplementary Fig. 2 for results concerning spindles detected simultaneously in MD and ANT and spindles detected in MD only.

## ANT spindles lock to early SO phases

So far, we established that both, neocortical SOs and spindles are preceded by corresponding activity in the ANT. These results also imply that the properties of local SO-spindle interactions, which are thought to shape memory consolidation[12], may differ between thalamic and neocortical sites, to allow for a concomitant thalamocortical coordination of SOs and spindles. To address this, we isolated SO-spindles complexes in ANT as well as scalp recordings and extracted the onset latencies of spindles relative to their corresponding SO down-states within both regions, respectively.

In line with previous findings[16,25], occurrence probabilities of neocortical spindles with respect to neocortical SOs (Fig. 6a), indicated that spindles started specifically at the SO down-to-up transition (0.05–0.15 s; $p < 0.0001$; peak: 0.1 s; corrected for multiple comparisons across time). In contrast, ANT spindles preferential emerged around ANT SO down states (−0.15 to 0.25 s; $p = 0.0012$; peak: 0 s; corrected for multiple comparisons across time. Fig. 6b). These diverging dynamics also became apparent when depicting the SO-locked (down-state) neocortical and ANT derived SO-spindle TFRs (Fig. 6c: locked to neocortical SO down state; Fig. 6d: locked to ANT SO down state). TFRs for both ANT and scalp-derived SO-spindle complexes exhibited characteristic power increases in the SO and sleep spindle band ($p < 0.05$, corrected for multiple comparisons across time and frequency). However, while neocortical spindles showed their typical power peaks well after the corresponding down-states[14], increases in the ANT sleep spindle band were initiated around the ANT SO down-states.

Finally, directly comparing the occurrence probabilities of thalamic and neocortical SO-spindle complexes affirmed that ANT spindles locked significantly earlier to SOs than neocortical spindles (−0.3 to −0.1, $p = 0.0039$; corrected for multiple comparisons across time; Fig. 6e). In a complementary analytical approach, we determined the preferred phase of SO-spindle modulation (i.e., SO-spindle coupling), for both ANT and neocortical events, respectively, by assessing the SO phases corresponding to the spindle onset latencies in each contact/scalp electrode. In 12/12 ANT contacts we found significant nonuniform distributions ($p < 0.05$, corrected for multiple comparisons using FDR; Rayleigh test, mean vector length: $0.41 \pm 0.03$) and a significant nonuniform distribution across contacts (Rayleigh $z = 9.33$, $p < 0.0001$), with spindles starting near the SO down state (corresponding to ± 180°; mean coupling direction: −175.22° ± 6.92°; see Fig. 6f). Similarly, we found in 7/8 scalp contacts a significant nonuniform distribution ($p < 0.05$, corrected for multiple comparisons using FDR; Rayleigh test, mean vector length: $0.27 \pm 0.02$). Again, a significant nonuniform distribution across participants was present (Rayleigh $z = 6.91$, $p < 0.0001$), with spindles preferentially starting at the SO down-to-up transition (mean coupling direction: −137.14° ± 7.58°; see Fig. 6f). Next, we tested whether the preferred coupling phases would vary systematically between ANT and neocortical areas, using the circular Watson–Williams test. Indeed, we found a significant difference in relation to the preferred phase of ANT and neocortical SO-spindle modulation ($F = 9.31$; $p = 0.0069$), corroborating the finding that spindles in the thalamus lock to earlier phases of the SO. Finally, we quantified the directional influence of SOs on

spindles in both ANT and neocortical data, using the PSI[26]. We found that SOs predicted sleep spindle activity within ANT and neocortex, respectively, as evidenced by a positive PSI (ANT: $0.0040 \pm 0.0017$; $t$-test against zero: $t = 2.37$, $p = 0.037$; neocortex: $0.0048 \pm 0.002$; $t$-test against zero: $t = 2.44$, $p = 0.044$; Fig. 6g).

## MD spindles lock to early SO phases

We have shown that neocortical SOs precede MD SOs, while MD spindles lead neocortical spindles. Accordingly, the coupling within MD-related SO-spindle complexes is expected to be shifted compared to neocortical SO-spindle complexes. Occurrence probabilities indicated that neocortical spindles started specifically at the down-to-up transition of neocortical SOs (0.1–0.3 s; $p = 0.003$; peak: 0.1 s; corrected for multiple comparisons across time; Fig. 7a). MD spindles, however, started preferentially briefly before the MD SO down-states (−0.35 to 0 s; $p < 0.0001$; peak: −0.05 s; Fig. 7b). Figure 7c, d show the neocortical and MD SO-spindle TFRs (Fig. 7c: locked to neocortical SO down-states; Fig. 7d: locked to MD SO down-state). TFRs for both scalp and MD-derived SO-spindle complexes featured power increases in the SO and sleep spindle band ($p < 0.05$, corrected for multiple comparisons across time and frequency). Neocortical spindles exhibited their typical power peaks after the corresponding down-states, while power peaks of MD sleep spindles started around MD SO down-states. Accordingly, the direct comparison of occurrence probabilities revealed that MD spindles locked significantly earlier to accompanying MD SOs as did neocortical spindles to neocortical SOs (−0.35 to 0.05, $p < 0.0001$; corrected for multiple comparisons across time; Fig. 7e).

Next, we determined the preferred phase of SO-spindle modulation for MD and neocortical events, respectively, by assessing the SO phases corresponding to the spindle onset latencies in each contact/scalp electrode. In 15/15 thalamic contacts, we found significant nonuniform distributions ($p < 0.05$, corrected for multiple comparisons using FDR; Rayleigh test, mean vector length: $0.42 \pm 0.03$). Also, across contacts the distribution was significantly nonuniform (Rayleigh $z = 12.10$, $p < 0.0001$), with spindles starting near the SO down-state (corresponding to ± 180°; mean coupling direction: 172.26° ± 6.67°; see Fig. 7f).

For scalp derived data, we found in 9/11 contacts a significant nonuniform distribution ($p < 0.05$, corrected for multiple comparisons using FDR; Rayleigh test, mean vector length: $0.21 \pm 0.02$). Again, a significant nonuniform distribution across participants was present (Rayleigh $z = 8.18$, $p < 0.0001$), with spindles preferably starting at the down-to-up transition of neocortical SOs (mean coupling direction: −138.43 ± 9.05°; see Fig. 7f). When directly comparing the preferred coupling phases between MD and neocortical areas, we found a significant difference (Watson–Williams test: $F = 17.33$; $p = 0.0003$), supporting the outcome that spindles in the MD start at earlier phases of the SO than neocortical spindles. We also quantified the directional influence of SOs on spindles in both MD and neocortical data, using the phase-slope index[26]. Again, we show that SOs predicted sleep spindle activity within MD and neocortex, respectively, as evidenced by a positive PSI (MD: $0.0016 \pm 0.0006$; $t$-test against zero: $t = 2.36$, $p = 0.032$; neocortex: $0.0028 \pm 0.002$; $t$-test against zero: $t = 2.23$, $p = 0.049$; Fig. 7g). Finally, we compared the features of ANT- and MD-derived SO-spindle complexes. While MD related sleep spindles had a tendency to lock to even earlier phases of MD related SOs (Fig. 7h), this difference was not significant (occurrence probabilities: $p = 0.11$; Fig. 7i: phase distribution: $F = 1.29$; $p = 0.25$). There was no difference in the SO-spindle coupling rate between ANT and MD ($t = 0.6057$, $p = 0.55$).

## Discussion

Our results suggest that anterior thalamic activity might play a fundamental role in the orchestration of cardinal sleep rhythms in humans. In particular, we found that SOs recorded in the human ANT

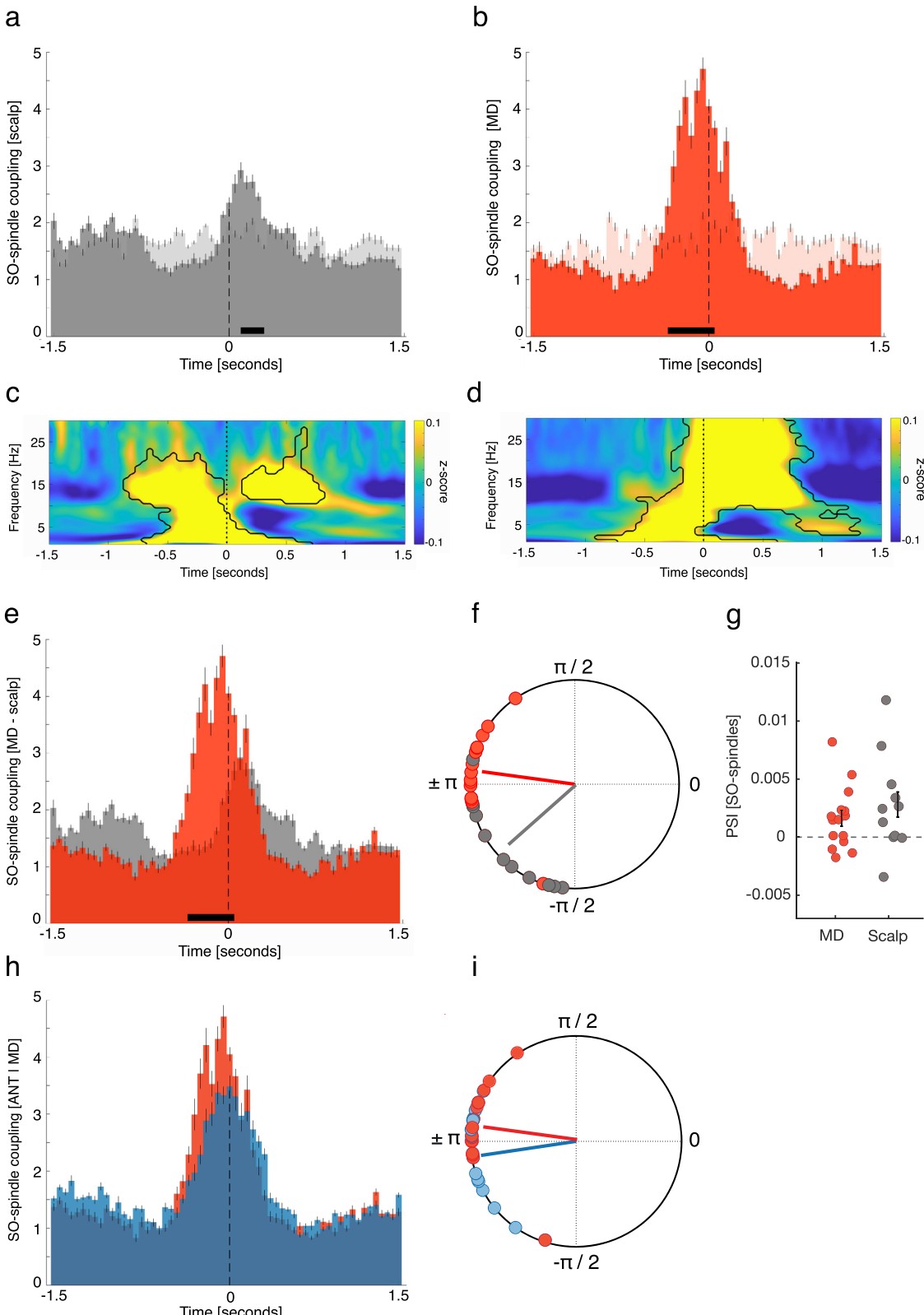

preceded their neocortical counterparts (average peak: −50 ms), in contrast to SOs recorded in the MD, which were led by neocortical SOs (average peak: +50 ms). Spindles recorded in ANT as well as MD preceded neocortical spindles (average peak: −100 ms). Finally, both in the ANT and the MD, thalamic spindles locked to early SO-phases, facilitating the prototypical synchrony between SOs and spindles in the neocortex.

Sleep oscillations have gained considerable interest over the last decade, mainly due their presumed role in facilitating the memory function of sleep[18]. SOs are thought to represent the time-giving pacemakers of memory consolidation and have been studied extensively in both rodent models and during human sleep[14,27]. Traditionally, they have been assumed to be solely generated within neocortical circuits[3,4]. However, recent animal studies cast doubt on this view by

**Fig. 7 | MD spindles lock to early SO phases. a** Occurrence probabilities of neocortical spindle onsets with respect to SO down-states (bin size = 50 ms), indicating that spindles started at the SO down-to-up transition (0.05 to 0.15 s; dependent-samples $t$-test, two-sided, $p < 0.0001$; peak: 0.1 s; corrected across time). **b** MD spindles started around SO down-states (−0.35 to 0 s; dependent-samples $t$-test, two-sided, $p < 0.0001$; peak: −0.05 s; corrected across time;). **c** Time–frequency representation (TFR) of neocortical SO-spindles ($N = 338.5 \pm 79.7$; locked to down-states), contrasted against event-free segments. Contour lines indicate clusters (dependent-samples $t$-test, two-sided: $p < 0.05$, corrected across time and frequency). **d** TFR of all MD SO-spindles (MD: $N = 488 \pm 53.3$; locked to down-states), contrasted against event-free segments. Contour lines indicate clusters (dependent-samples $t$-test, two-sided: $p < 0.05$, corrected across time and frequency). **e** Comparing the occurrence probabilities of MD (red) and neocortical SO-spindles (gray) revealed that MD spindles started at earlier SO phases (−0.35 to 0.05, dependent-samples $t$-test, two-sided: $p < 0.0001$; corrected across time). **f** Phases

of neocortical (gray) and MD (red) SO-spindle modulations. Neocortical spindles started at the SO down-to-up transition (phase of down-states corresponds to $\pm \pi$; mean coupling direction: $-138.43 \pm 9.05°$; Rayleigh test, one-sided: $z = 8.18$, $p < 0.0001$), while MD spindles started around the SO down-states (mean coupling direction: $172.26 \pm 6.67°$; Rayleigh test, one-sided: $z = 12.10$, $p < 0.0001$). The preferred phases of SO-spindle modulation differed between MD and neocortical sites (Watson–Williams test, two-sided: $F = 17.33$; $p = 0.0003$). **g** Directional SO-spindle coupling obtained by the phase-slope index (PSI; mean ± SEM) for MD (red) and neocortical (gray) SO-spindles. SO-phases predicted spindle amplitudes in MD ($t$-test against zero, two-sided: $p = 0.032$) and neocortical SO-spindles ($t$-test against zero, two-sided: $p = 0.049$). **h**, **i** Directly comparing ANT- (blue) and MD derived SO-spindles (red) did not yield a significant difference (**h**) occurrence probabilities (dependent-samples $t$-test, two-sided: $p = 0.11$, corrected across time); **i** phase distribution (Watson–Williams test, two-sided: $F = 1.29$; $p = 0.25$).

---

identifying essential thalamic contributions for SO generation and coordination[5–10]. Thalamic neurons have been shown to exhibit intrinsic, rhythmic up- and down-states in isolated conditions (i.e., without neocortical inputs[5,28]), while phasic burst-firing of thalamocortical neurons at the onset of up-states often precedes the firing of neocortical neurons by 20–50 ms[7,29]. Optogenetic stimulation of thalamocortical neurons efficiently initiates neocortical up-states[6,30]. In addition, thalamic activity terminates SO up-states synchronously across neocortical areas, thus mediating a well-controlled down-state transition[31]. In sum, these findings suggest that thalamic activity exerts a controlling influence on neocortical SOs, by coordinating up- and down-state dynamics in neocortical circuits. However, the generalization of thalamic contributions, and specifically of distinct thalamic nuclei, to sleep oscillations has been shown to be challenging across species (e.g.,[32,33]), prompting their direct assessment during human sleep.

To investigate potential contributions of the ANT to neocortical SO coordination, we detected SOs during NREM sleep in intracranial contacts located in the ANT as well as scalp EEG. We show that the ANT not only exhibits a slow oscillatory rhythm (note that the ANT signal is likely to be of local origin due to a bipolar montage), but, importantly, that ANT SOs precede their neocortical counterparts. This might indicate that the ANT guides SO activity in the neocortex. Focusing on paired events only (i.e., time windows where both neocortical and ANT SOs were present), we found that ANT SO phases significantly lead neocortical SO phases. Notably, this pattern of results was specific to ANT recordings. SOs in the MD did not precede but rather followed neocortical SOs, which is in line with a previous report of neocortical SOs leading SOs in the human pulvinar[13]. Our findings thus highlight both a specific role of the ANT for SO dynamics and, more generally, the functional diversity of specific thalamic nuclei in humans[34], also with regards to sleep oscillations potentially originating there. Since scalp EEG recordings provide an indirect measure of the underlying cortical sources, one might ask whether a delay in cortical SOs relative to ANT SOs might be driven by effects of interjacent tissue (e.g., the skull) delaying the timing of the signal picked up at frontal scalp EEG channels. To mitigate these concerns, we show, in an independent data set, that intracranially recorded SOs in the frontal lobe exhibit no temporal delay with regards to SOs simultaneously recorded by frontal scalp EEG (see Supplementary Fig. 3). However, we acknowledge that a bias in the relative timing between ANT and scalp SOs cannot be completely excluded due to technical differences in recording techniques (intracranial vs. scalp EEG).

Another concern might arise considering volume conduction contaminating the thalamic signal. We used a bipolar montage for thalamic contacts to ensure a local origin of the signal[35–37]. However, there might still be a possibility that the signal recorded from these contacts is contaminated by volume conducted activity from adjacent brain tissue outside the thalamus. We address this issue in a dedicated

control analysis (see Supplementary Fig. 4) and conclude that none of these factors is likely to systematically bias our findings. The dissociation of SO timing between ANT and MD SOs, which are anatomical neighbors, also speaks against a global, volume conducted source driving the effects. In addition, our data were recorded in epilepsy patients in whom anti-epileptic medication and epileptiform events may affect sleep architecture and sleep-related oscillations[38–41], warranting caution when generalizing to healthy population. However, our patient population was heterogeneous in terms of medication, age, epilepsy form and affected cortical area. Furthermore, sleep architecture and SO- and spindle quantity and densities were within normal ranges of healthy participants (see Supplementary Tables 3 and 4[42,43]), making it unlikely that a single aspect had a systematic impact on our results. In addition, while only seizure-free nights were included in the analysis, our strict artifact rejection procedure minimized the impact of epileptiform activity on our findings.

What makes the ANT a favorable candidate for sculpting neocortical SOs? The ANT is a key node in the limbic circuit[44] and functionally relevant for human memory[20,22]. It has extensive connections with the anterior cingulate, orbitofrontal cortex and the hippocampus[45,46]. Notably, medial prefrontal regions, such as the orbitofrontal and the cingulate cortex have been identified as neocortical hot spots for the emergence of SOs during human NREM sleep[47,48]. In addition, work in rodent models has shown that midline and anterior thalamic nuclei are likely candidates to influence global neocortical activity during sleep due to their widespread projections[7,31]. Hence, the dense connectivity of the ANT with medial prefrontal regions and within the limbic circuit sets the stage for ANT activity to shape neocortical SOs and coordinate sleep oscillations between regions implicated in memory consolidation.

We here show ANT SOs leading neocortical SOs on the basis of approximately 30% of neocortical SOs that were coupled to ANT SOs. Neocortical SOs that were not coupled to the ANT may as well have been initiated in the neocortex, in line with traditional accounts of the origin of SOs[3,4]. Future research needs to investigate whether ANT coordinated SOs are functionally distinct from neocortically coordinated SOs, for example with respect to memory reprocessing during sleep. The precise coupling between SOs and thalamic spindles might be of relevance here, given that the precision of the coupling between SOs and spindles has been shown to be instrumental for memory consolidation to unfold (e.g. refs. 12, 25). Due to its prominent location within the limbic circuit and direct projections to the hippocampal formation[21], the ANT could also be important for integrating hippocampal ripples into SO-spindle complexes, facilitating the full triple-coupling of sleep oscillations that is assumed to mediate the memory function of sleep[18]. Hence, our work represents a starting point for elucidating the distinct role of the ANT in coordinating sleep oscillations and memory consolidation.

We also found clear evidence for spindles in ANT and MD. Spindles emerge in recurrent loops between the thalamic nucleus reticularis (TRN) and thalamocortical neurons, which in turn forward them to the neocortex and hippocampus[23]. There have been conflicting findings across animal species about whether the TRN innervates the ANT and, hence, whether or not spindles are present in the ANT[32,33,49–51]. Our results provide a clear picture in humans: spindles are not only present both in the ANT and MD but also precede neocortical spindles, indicating an involvement of both nuclei in spindle generation and their projection to neocortical areas. In line with our interpretation, optogenetic stimulation of the rodent anterodorsal TRN, which innervates the ANT[33,52], has been shown to initiate spindles in the neocortex and hippocampus[53], indicating that the ANT might synchronize key areas of memory consolidation[54]. With regards to the MD it has been found that its neurons exhibit increased firing during spindle events in rodents[55]. In humans, reduced MD volume in persons with schizophrenia is associated with dampened neocortical spindle activity[56], further pointing towards a role of the MD in the thalamocortical spindle transfer. In line with previous work on spindles in posterior thalamic nuclei[57], we identify local and non-local spindles in ANT and MD. With respect to the neural circuitry generating spindles in ANT and MD, one could assume recurrent loops between TRN and thalamocortical neurons, similar to what has been shown in animals[11,23]. Extrapolating these findings from animals to humans needs to be done with caution, however, since result from animal models concerning TRN innervation of thalamic nuclei is not always univocal[32,33]. In general, it is assumed that TRN is organized in sectors that topographically project to thalamic nuclei[33], which can explain why local spindles emerge in both MD and ANT. However, the mapping of TRN sections to thalamic nuclei is not one-to-one[58]: for example, some TRN sectors projecting to the MD also project to ANT[59–61]. Such overlapping projections seem plausible when considering that MD and ANT thalamocortical neurons share common targets, for example in the prefrontal cortex.

In sum, our results extend previous research identifying sleep spindles in other human thalamic nuclei[13,57], by showing that sleep spindles in the ANT and MD precede neocortical spindles during human sleep, indicating a role of both nuclei in thalamocortical spindle projections.

Finally, we set out to characterize the properties of SO-spindle interactions within the ANT and the MD. Spindles tend to nest within SO up-states in scalp EEG recordings[12,14,25]. Importantly, the precise timing of neocortical spindle peaks with respect to the SO up-states has been shown to be tightly linked to the behavioral expressions of memory consolidation[12,25], putatively by igniting structural changes in neocortical sites[62]. Hence, the characteristics of SO-spindle interactions in the ANT might highlight how the thalamus simultaneously groups SOs and spindles in the neocortex, leading to the well-known oscillatory nesting (i.e., spindles nesting towards SO up-states). We found that spindles in the ANT were initiated around the thalamic SO down-states, in contrast to neocortical spindles that preferentially started later, at the neocortical SO down-to-up transition transition (leading to the well-known nesting of neocortical spindles towards the up-states)[1,14].

Thus, ANT spindles started at significantly earlier ANT SO phases as compared to neocortical SO-spindle interactions. Notably, this divergent property of SO-spindle coordination in the ANT is a consequence of the observed delays between ANT and cortical sleep oscillations. It allows spindles, putatively generated in the ANT, to arrive during SO up-states on neocortical sites, generating the prototypical SO-spindle coupling found in scalp EEG. Specifically, while we show that both ANT SOs and spindles preceded their neocortical counterparts, the time lag between ANT and neocortical spindles exceeded the time lag between ANT and neocortical SOs (100 ms vs. 50 ms). Hence, to enable the prototypical SO-spindle modulation in

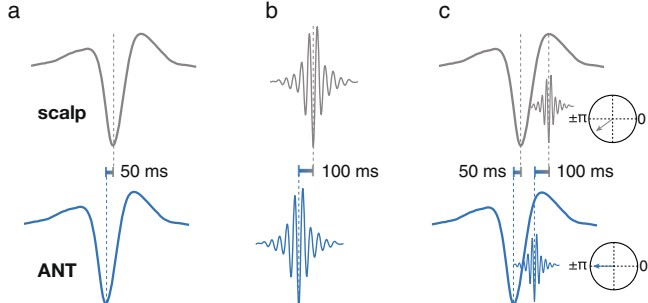

**Fig. 8 | Schematic overview of the interplay between sleep oscillations recorded in the ANT and on the scalp. a** ANT SOs (bottom, blue) preferentially peak on average ~50 ms before the emergence of neocortical SOs (top, gray). **b** ANT spindles (bottom, blue) peak on average 100 ms before neocortical spindles (top, gray). **c** ANT spindles couple to earlier ANT SO phases (bottom, blue) than neocortical spindles couple to neocortical SOs (top, gray). While neocortical oscillations show the prototypical SO-spindle modulation (i.e., nesting of spindles towards the SO up-state), spindles in the ANT start closer to the ANT SO down-state. The insets illustrate the difference in the phase locking of spindle start to their respective SO (the phase of SO down-states corresponds to ± π).

neocortical circuits (i.e., nesting of spindles towards the SO up-states) under the assumption that both graph-elements in the neocortex are governed by the ANT, it is inevitable that spindles start at relatively earlier phases of the SO in the ANT as compared to spindles of neocortical SO-spindle complexes. Without such a shifted coordination on the thalamic level, cortical SO-spindle coupling would be systematically misaligned, with spindles regularly peaking after SO up-states (see Fig. 8 for a schematic illustration).

We found a similar pattern of SO-spindle interactions within the MD, with spindles starting at early phases of the SO. Interestingly, even though not significant, spindles in the MD seem to start even earlier with respect to SOs as it is the case in the ANT. Again, in order to project spindles to the neocortex that nest in neocortical up-states in a timely manner, a shifted coordination of SOs and spindles in the MD is necessary (i.e., spindles starting at early SO phases). This seems to be even more relevant in the case of the MD, where SOs were preceded (and potentially driven) by neocortical SOs, while spindles preceded neocortical spindles. Hence, for MD spindles to arrive at the neocortex during the presence of an SO up-state, it is inevitable to start at an early MD SO phase. The tight coupling of thalamic spindles to thalamic down-states is in line with previous work showing a very similar temporal relationship in the human pulvinar[13]. These results indicate that the hyperpolarization during thalamic down-states is required to trigger the neuronal cascade underlying spindle generation.

Taken together, our findings provide evidence for the ANT as a major hub for coordinating the cardinal NREM sleep-related oscillations. In the neocortex, both spindles and SOs were led by ANT-related activity, while the interplay of local ANT SO-spindles was tuned in a way that allows for neocortical spindles to nest within SO up-states (ANT spindles starting at early ANT SO phases, see Fig. 8). Importantly, this exact coordination is thought to be instrumental for the consolidation of memories[1]. Hence, the ANT might represent a key player in governing memory re-processing during human NREM sleep.

Due to its projections to the medial prefrontal cortex and the hippocampus, the ANT has already been implicated in facilitating memory functioning and spatial navigation during wake, putatively by facilitating hippocampal-neocortical interactions[22,54,63–65]. Notably, the same hippocampal-cortical dialog is assumed to underlay sleep-related memory consolidation, facilitated by a precise coupling of hippocampal ripples, neocortical SOs and thalamic spindles[1]. The standard model suggests that neocortical SOs are imposed on a rather passive thalamus, where they initiate spindles. Spindles in turn are thought to synchronize

hippocampal ripples and reactivated memory information[16,66]. Through this coordination the reactivated memory information is suggested to arrive at the neocortex during periods of high plasticity (i.e., during the presence of SO up-states and spindles)[62].

Our data critically extend these theoretical considerations, by spotlighting the human thalamus, and in specific the ANT, as a putative active agent in interfacing sleep-related oscillations between brain regions and thereby, potentially facilitating memory consolidation. While our current data remain agnostic with regards to hippocampal activity, it has been shown in a single patient study, that hippocampal spindles are aligned with SOs in the ANT[67]. Hence, ANT-related oscillations might not only migrate to the neocortex but likewise to the hippocampus, where spindles are known to govern ripples and associated memory reactivation[68]. Taken together, ANT-related activity might critically contribute to the triple-coupling of sleep oscillations and thus, the memory function of sleep. Future work relating the association of ANT-neocortical interactions with the behavioral expressions of memory consolidation will elucidate the memory function of the ANT and further characterize the functionally diverse human thalamus.

## Methods

### Participants
Intracranial and scalp EEG were simultaneously recorded from patients with bilateral depth electrodes implanted in the thalamus for deep brain stimulation (DBS) therapy of pharmaco-resistant epilepsy. Data was recorded at the Epilepsy Center, Department of Neurology, Ludwig–Maximilian Universität (Munich, Germany), while DBS leads were externalized post-surgery, prior to their connection to the pulse generator located in a subcutaneous infraclavicular or abdominal pouch[69]. The availability of direct iEEG recordings from the human thalamus is highly limited due to (i) the rarity of patients being treated with thalamic DBS, (ii) access to thalamic iEEG in these patients (post-surgery externalization of DBS leads is performed only in a fraction of these patients), and (iii) the limited amount of time for externalization (after surgery and before connecting the leads to the pulse generator).

A total of 13 patients participated. The data of five patients were discarded due to excessive epileptic activity. Out of the remaining eight patients (mean age: $38.65 \pm 3.59$; five female), six patients contributed single full-night recordings, one patient contributed two consecutive full-night recordings and one patient contributed three consecutive full-night recordings. In sum, 11 full-night recordings entered the analyses of the present study. All patients received anticonvulsive medication (see Supplementary Tables 7 and 8 for each participant's drug regimen at the time of recordings and their influence on sleep architecture). Informed consent was obtained from all patients. The study was approved by the ethics committee of the Medical Faculty of the Ludwig–Maximilian Universität.

### Intracranial data acquisition and electrode localization
iEEG data were recorded from thalamic depth electrodes that each had four intracranial electrode contacts (platinum–iridium contacts, 1.5 mm wide with 1.5 mm edge-to-edge distance; Medtronic 3387). Data were recorded using XLTEK Neuroworks software (Natus Medical, San Carlos, CA, USA) and an XLTEK EMU128FS amplifier, with voltages referenced online to electrode CPz (250 Hz sampling rate in seven patients; 200 Hz sampling rate in one patient).

The ANT in both hemispheres is the clinically-relevant implantation target for DBS therapy of epilepsy[70]. Due their small and high inter- and intraindividual variable size and the transventricular implantation trajectory, a subset of the electrode contacts ends up outside the ANT, frequently in the mediodorsal thalamus (see Fig. 1a).

The locations of the electrode contacts were estimated using the Lead-DBS toolbox[71]. First, the post-operative CT scan was co-registered to the pre-operative T1-weighted MRI, as implemented in the

Advanced Normalization Tools[72]. The scans were then spatially normalized to MNI space, based on the pre-operative T1 image using the unified segmentation approach as implemented in SPM12[73]. Next, the trajectories of the DBS electrodes and positions of the electrode contacts were reconstructed based on the post-operative CT scan. The positions of the electrode contacts where then visually confirmed using the DISTAL atlas (as implemented in the Lead-DBS toolbox)[74]. To mitigate effects of volume conduction on our analyses[75], the iEEG recordings were then re-referenced to their immediate neighboring contact (bipolar montage; see Supplementary Fig. 4 for an estimation of the potential influence of remaining volume conducted SOs on the main results). A bipolar pair of contacts was considered to be in the ANT if both or at least one contact was localized to the ANT, but no contact was localized in the MD (see Supplementary Fig. 5 for a comparison of full ANT contacts and single ANT contacts with regards to occurrence probabilities of SOs, spindles and SO-spindles). A bipolar pair of contacts was considered to be in the MD if both or at least one contact was localized to the MD, but no contact was localized in the ANT. This procedure yielded 20 thalamic contacts (seven patients) in the ANT, and 24 contacts (eight patients) in the MD (see Supplementary Fig. 6 for a comparison of left and right ANT / MD contacts with regards to occurrence probabilities of SOs and spindles).

To investigate whether the signal picked up at the scalp level might have been delayed by interjacent tissue (e.g., the skull), we assessed SO coupling between frontal scalp electrodes (electrode FP1) and frontal intracranial contacts in an independent data set consisting of three pre-surgical epilepsy patients (for details see Supplementary Fig. 3).

### Scalp EEG acquisition
Scalp EEG was simultaneously recorded with iEEG via a common amplifier. Scalp EEG electrodes were placed on the accessible scalp according to the international 10–20 system for electrode placement. The sampling rate of the scalp EEG was identical to the sampling rate of the iEEG. Scalp EEG was recorded from 20 (two patients), 22 (three patients), 24 (one patient) or 36 electrodes (two patients). Scalp EEG was online referenced to CPz. Scalp EEG data were re-referenced offline against the mean of all available artifact-free scalp EEG electrodes.

### Sleep staging
All available scalp electrodes were low-pass filtered at 40 Hz and downsampled to 200 Hz (if necessary). Sleep staging was carried out according to standard criteria[76]. For subsequent analyses, sleep stages N2 and N3 were collapsed and referred to as non-rapid eye movement (NREM) sleep.

### Data pre-processing
iEEG and scalp EEG were manually inspected to discard noisy or artifact-contaminated channels and data segments (see below for detection of epileptic activity). All data were downsampled to 200 Hz, demeaned and de-trended. The recordings were then filtered using a 150 Hz Butterworth low-pass filter and a two Butterworth band-stop filters (to attenuate line noise; 48–52 Hz).

**Interictal epileptic discharge (IED) detection.** In general, only seizure-free nights were included in the analysis. Epileptic activity in scalp and thalamic recordings was semi-automatically detected. First, interictal epileptic discharges were detected automatically by high- and low-pass filtering the data below 20 and above 80 Hz and z-scoring the data. Subsequently, data segments with amplitudes exceeding the mean signal by 6 SDs for less than 100 ms were identified. Second, iEEG and scalp EEG data were visually inspected for epileptic activity by two investigators, independently. Data segments comprising epileptic events at any given channel were discarded (13.91 ± 2.41% of all sleep

data). Four thalamic contacts (2 ANT contacts, 2 MD contacts) and on average 7.5 ± 0.92 scalp electrodes were discarded.

## Event detection

SOs and sleep spindles were identified for each patient, based on established detection algorithms[16,25].

**SO detection.** Data were filtered between 0.3–2 Hz (two-pass FIR bandpass filter, order = three cycles of the low frequency cut-off). Only movement-free data (as determined during sleep scoring) from NREM sleep stages two and three were considered. All zero-crossings were determined in the filtered signal of each channel (iEEG and scalp EEG, respectively), and event duration was determined for SO candidates (that is, down-states followed by up-states) as time between two successive positive- to-negative zero- crossings. Events that met the SO duration criteria (minimum of 0.8 and maximum of 2 s) and exceeded the mean amplitude of all detected events by 1.25 SD entered the analysis. For subsequent time-locked analyses (ERPs, time–frequency representations, etc.), 5-s-long segments centered on the down state (± 2.5 s) were extracted from the unfiltered raw signal.

**Spindle detection.** Data were filtered between 11–16 Hz (two-pass FIR bandpass filter, order = three cycles of the low-frequency cut-off). Again, only artifact-free data from NREM sleep stages 2 and 3 were used for event detection. The root mean square (RMS) signal was calculated for the filtered signal at each channel using a moving average of 200 ms, and spindles that exceeded the mean amplitude of all detected events by 1.5 SD entered the analysis. Whenever the signal exceeded this threshold for more than 0.5 s but less than 3 s (duration criteria), a spindle event was detected. Spindle onsets were defined as the points in time when the signal initially crossed the threshold. Epochs time-locked to the maximally negative spindle trough (−2.5 to +2.5 s) were extracted from the unfiltered raw signal for subsequent time-locked analyses.

**SO-spindle complexes.** To isolate SO-spindle complexes in scalp and thalamic recordings, we determined for all SOs whether a spindle was detected following the SO (SO down-state + 750 ms). Again, SO-spindle events were extracted (−2.5 to +2.5 s with regards to the SO down-state) from the raw signal for subsequent time-locked analyses.

**Paired events.** To identify paired SO events (scalp & thalamus), we determined for all scalp detected SOs whether corresponding SOs were detectable in thalamic recordings within a time window of ±750 ms. Data segments locked to the scalp detected down-states (± 2.5 s) were extracted. Likewise, paired sleep spindles were isolated by determining for all scalp detected sleep spindles whether related spindles emerged (±750 ms) in thalamic recordings. Data segments locked to the scalp detected maximally negative spindle peak (±2.5 s) were extracted.

**Electrode selection [scalp EEG].** As outlined above, all electrodes exhibiting significant epileptic activity were discarded to ensure that our results were not influenced by epileptiform activity. This conservative procedure together with restricted access to the scalp due to post-surgical wound care impeded us from using identical scalp EEG electrodes across patients to delineate thalamocortical interactions during sleep (e.g., electrode Fz for SO-related and electrode Cz for sleep spindle-related analyses)[16]. Instead, we pursued a data-driven and individualized approach. For SO-related analyses, we selected for each participant the frontal scalp electrode that exhibited the highest number of associated thalamic SOs within a time of ±750 ms. This procedure was accomplished independently for data stemming from the ANT and the MD. The respective scalp electrodes (one related to the ANT, one related to the MD) were used for all subsequent SO-

related analyses (see Supplementary Table 1 for an overview of the selected scalp electrode; see Supplementary Fig. 7 for ANT SO occurrence probabilities relative to scalp electrode Fz; see Supplementary Fig. 8 for ANT SO occurrence probabilities relative to occipital scalp electrodes). Similarly, for sleep spindle-related analyses, we selected the frontal, central or parietal scalp electrode that exhibited the highest number of associated thalamic spindles (ANT or MD, respectively; see Supplementary Table 2 for on overview of the selected scalp electrodes; see Supplementary Fig. 9 for spindle–spindle occurrence probabilities, where scalp spindles were exclusively derived from frontal or parietal electrodes; for slow and fast spindle-related results see Supplementary Figs. 10, 11 and Supplementary Table 9; for a single subject spindle ERP see Supplementary Fig. 12). The respective scalp electrodes (one related to the ANT, one related to the MD) were subsequently used for all sleep spindle-related analyses. Data from scalp electrodes were used as a proxy of neocortical activity.

**Event-free segments.** For statistical comparisons, we extracted 5-s-long intervals during NREM sleep, which did not exhibit any SO, spindle or SO-spindle event, respectively. In each participant and electrode, the number of control events corresponded to the number of prior detected events of interest. Event-free events were only drawn from time window starting 5 min before and ending 5 min after the corresponding oscillatory event.

## Analyses

**Temporal relation between sleep oscillations.** To assess the temporal relationship between scalp- and thalamus-derived SO events, we created peri-event time histograms (bin size = 50 ms) where scalp detected SO down-states served as seed (time = 0), while the targets (thalamic SO down-states) are depicted relative to the seed. The resulting histogram was normalized by the total number of detected scalp SOs (multiplied by 100) per participant and electrode site. Similarly, for sleep spindles, peri-event histograms (bin size = 50 ms) of thalamic sleep spindle peaks (maximally negative amplitude) in relation to scalp sleep spindles peaks (maximally negative amplitude: ± 1.5 s) were created. The resulting histogram was normalized by the total number of detected scalp spindles (multiplied by 100). To determine the interplay of SOs and sleep spindles (separately within scalp and thalamic recording), a peri-event histogram of sleep spindles (onsets) around SO down-states (±1.5 s) was created. The resulting histogram was normalized by the total number of detected spindles (multiplied by 100). The same procedure was applied to spindle-free control segments.

**Phase relationship between sleep oscillations.** To estimate the phase-relationship between scalp and thalamic SOs, data epochs including paired events were filtered in the SO range (0.3–2 Hz, two-pass Butterworth bandpass filter). As described above, these segments included both scalp and thalamic SOs, and were time-locked to the down-states of scalp detected SOs. A Hilbert transform was applied to the data and the instantaneous phase angle of the thalamic recording at the time of the scalp SO down-state was extracted. Each participant's preferred phase of SO–SO coupling was then obtained by taking the circular mean of all individual events' preferred phases.

For the analysis of SO-spindle coupling, we filtered the SO-spindle data (locked to spindle onsets) in the SO range (0.3–2 Hz, two-pass Butterworth bandpass filter), applied a Hilbert transform and extracted the instantaneous phase angle. Next, we isolated the SO phase angle at the time of spindle onsets. Each participant's preferred SO phase at spindle onsets was obtained by taking the circular mean of all individual events' preferred phases.

**Spindle-related across-channel power correlations.** We further estimated the temporal relationship of thalamic and scalp-detected

spindles using channel-to-channel correlations of oscillatory power in the sleep spindle band. To this end, a Hanning window was applied to paired sleep spindles segments (locked to scalp detected maximally negative amplitude) and power values between 11 and 16 Hz were extracted for each time-bin. Power values were *z*-scored per channel and averaged. Subsequently, power values derived from thalamic channels were correlated with scalp-related power values at each time-point (Spearman correlation), resulting in a 2d correlation map per contact pair. Finally, the correlation maps were normalized using the Fisher z-transform. The same procedure was applied to spindle-free control segments. Off-diagonal correlations indicate a time lag between spindle power in scalp and thalamic channels.

**Phase slope index.** We assessed whether SOs drive activity in the sleep spindle range or vice versa in NREM sleep data segments comprising SO-spindle complexes, separately for thalamic and scalp recordings. The cross-frequency phase-slope index[24], was calculated for each contact between the signal and the signal filtered in the sleep spindle range (11–16 Hz). After applying a Hanning window and extracting the complex Fourier coefficients, all SO frequencies <2 Hz were considered. In this context, positive values indicate SOs driving sleep spindle activity, while negative values indicate sleep spindles driving SOs. The obtained data distributions were tested against zero, using paired samples t-tests.

**Time–frequency representations.** Time–frequency analyses of SO, sleep spindle and SO-spindle segments were performed using FieldTrip[77]. Frequency decomposition of the data was achieved using Fourier analysis based on sliding time windows (moving forward in 50 ms increments). The window length was set to five cycles of a given frequency (frequency range: 1–30 Hz in 1 Hz steps). The windowed data segments were multiplied with a Hanning taper before Fourier analysis [−2.5 to 2.5 s]. The longer time segments were chosen to allow for resolving low-frequency activity within the time windows of interest [−1.5 to 1.5 s] and avoid edge artifacts. Resulting power values were z-scored across time. The same procedure was applied to event-free control segments.

**Statistics.** Statistical analyses were performed at the individual electrode/contact level (fixed-effects analysis), considering all electrodes/contacts that were eligible based on our criteria (e.g., artifact-free, etc.). Unless stated otherwise, we used cluster-based permutation tests to correct for multiple comparisons as implemented in FieldTrip[77]. A dependent-samples *t*-test was used at the sample level and values were thresholded at $p = 0.05$ (1000 randomizations). The sum of all *t*-values in cluster served as cluster statistic and Monte Carlo simulations were used to calculate the cluster *p*-value (alpha = 0.05, two-tailed) under the permutation distribution. The input data were either occurrence probabilities across time (e.g., Fig. 2a), time × frequency values (e.g., Fig. 2c) or time × time correlation maps (e.g., Fig. 4b) which were tested against corresponding data stemming from event-free events. In case of between area comparisons (e.g., Fig. 3g) independent samples *t*-tests (cluster corrected) were employed to conform with the varying number of observation units (i.e., contacts).

For circular statistics, the phase distributions (within or across participants) were tested against uniformity using the Rayleigh test (CircStat toolbox[78]). When directly comparing phase distributions across recording sites, the circular Watson–Williams test was used[78].

**Reporting summary**
Further information on research design is available in the Nature Research Reporting Summary linked to this article.

## Data availability
The raw data are available under restricted access for data privacy laws, access can be obtained by sending a reasonable request (thomas.schreiner@psy.lmu.de). Source data are provided with this paper.

## Code availability
Data pre-processing and analysis were performed using MATLAB 2020a and publicly available toolboxes (i.e.; electrode contact localization: LEAD-DBS v.1.6.3, https://github.com/netstim/leaddbs; SPM12, https://github.com/spm/spm12; scalp and intracranial EEG analysis: Fieldtrip Toolbox, v.09/01/2020, http://www.fieldtriptoolbox.org/; CircStat Toolbox, v.1, https://github.com/circstat/circstat-matlab). All scripts used to run the analysis are also available from the authors upon request (thomas.schreiner@psy.lmu.de).

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

## Acknowledgements

This work was supported by the European Research Council (https://erc.europa.eu/, Starting Grant 802681 awarded to TSt). We are indebted to all patients who volunteered to participate in this study. We thank the staff and physicians at the Epilepsy Center, Department of Neurology, Ludwig Maximilians University, Munich for assistance.

## Author contributions

Conceptualization: T.S., T.St.; Resources: E.K., S.N., J.H.M.; Formal Analysis: T.S., T.St.; Funding acquisition: T.St.; Writing – original draft: T.S., T.St.; Writing– review & editing: T.S., T.St., E.K., S.N.

## Funding

## Competing interests

The authors declare no competing interests.
