## [Peer Review File · Nature Communications]

The human thalamus orchestrates neocortical oscillations during NREM sleep.REVIEWER COMMENTS

Reviewer #1 (Remarks to the Author):

The authors analyzed the dynamics of slow oscillations (SO) and spindles during NonREM sleep which were concurrently recorded from the anterior (ANT) and mediodorsal thalamus (MD) using intracranial electrodes, and from frontal/frontotemporal and parietal scalp EEG electrodes. SOs identified in the ANT tended to precede SOs occurring in the EEG (over frontal cortex) whereas SOs identified in the MD tended to follow those identified in the EEG (mainly over parietal cortex). Spindles detected in thalamic recordings tended to precede those at respective EEG sites with, the delay of EEG spindles relative to thalamic spindles being greater than that for respective SOs, i.e., thalamic spindles tended to occur at an earlier phase of the respective SOs (identified in the same thalamic recordings) than EEG spindles which – as shown in many previous studies - occurred during the SO upstate of the EEG SOs at frontal and parietal sites.

These are overall exciting data that potentially provide highly valuable insights - in humans (!) - into the regulation of the sleep oscillations that are commonly thought to mediate memory systems consolidation during sleep. The methods are overall sound and the manuscript is written in a comprehensible manner. Against this backdrop I think this manuscript is clearly attractive for a broad readership. Nevertheless, a careful revision is necessary before publication can be recommended.

1) The most important result of the study is that, contrary to the widely held view of a cortical origin of SOs, SOs recorded from ANT precede SOs recorded from frontal/frontotemporal EEG electrodes by on average about 50 ms (as judged from Fig. 2), i.e., a temporal relationship suggesting that ANT SOs drive frontocortical SOs. The lead of ANT SOs is small but, physiologically plausible and highly relevant as to possible cause-effect relationships. The more so the authors are in need to convincingly exclude non-physiological/technical confounds. First, scalp EEG recordings differ from intracranial EEG recordings in that the signal picked up by the electrode, is filtered by skull, tissues etc. which altogether act as low pass filter that may well delay the measured SO negative peak amplitude in the EEG by 50 ms or so, with reference to intracranial recording. Second, the authors argue (line 508) that in order “... to mitigate effects of volume conduction on our analyses, the iEEG recordings were then re-referenced to their immediate neighboring contact (bipolar montage)...” However, the bipolar montage does not entirely exclude that ANT SOs partially represent volume conducted SOs invading ANT from other sites, including anterior neocortex. (To obtain clues as to a possible role of volume conductance, the authors might check for the occurrence of polarity reversals in the SO signal in recordings using a common reference for EEG and iEEG. Events with polarity reversal are more likely reflecting neocortical SOs, and might then be used to estimate potential technical delays in SO peaks between EEG and bipolar iEEG recordings). In sum, these possible technical confounds need to be carefully and convincingly addressed in the Discussion.

2) Although spindles are generally of more local nature than SOs, basically similar questions arise for spindles detected in ANT and MD: They may represent spindle originating from the respective thalamic region but, volume conductance cannot be excluded. In this regard, separate analyses of spindles that

do and do not cooccur in phase in ANT and MD recordings (the later representing localized thalamic spindles) might provide further insights into a possible “guiding” role of ANT in organizing the SO-spindle interplay.

3) As judged from Fig 6 and 7, spindles recorded in ANT and MD appear to cover the downstate of the SOs recorded in the same region. Assuming that SO downstates represent synchronized membrane hyperpolarizations, it is difficult to understand that SOs and spindles reflect potential activity in the same cells circuits. The excitable troughs of the spindle oscillations are typically accompanied by bursts of spikes considered to be driven by the depolarizing phase of the SO upstate. Is there a plausible mechanism that would explain the occurrence of a spindle in a strongly hyperpolarized network?

Minor points:

- Abstract/Results/Discussion should provide average durations of the time lags between SO/spindles between thalamic and EEG recordings.
- Line 43, studies of thalamic SOs are not so “recent”.
- Please, add reference to the table of patient data in the beginning of Results section
- Line 120 “...time locked to SO down states”, should be phrased more precisely “...time locked to the SO down state peaks”
- Reporting results of EEG SOs and spindles the authors should add where the recordings were performed (F/FT for SOs, P for spindles).
- The vertical zero line in the TF plots (e.g. Fig. 3e) is difficult to see
- Line 189 “...clusters: -2.5 to -0.05 s” 0.25?
- Fig 4a, there is a second significant peak on the positive side not mentioned in the text.
- Figure 4d shows grand average for spindles. Please indicate the n for these averages. I am surprised to see that several cycles of the spindle oscillations are preserved in this grand average although the frequency of individual spindles probably varies (to a certain degree) within and across the patients, such that the discrete oscillations are typically smeared in grand averages.
- Spindle-SO coupling was performed with reference to the spindle onset. I didn’t find a definition of the spindle onset. Please, add.
- Line 313 “..power peaks of MD sleep spindles started around MD..” I can’t see this in Fig 7 where power is increased distinctly before.
- Line 344 “Our results demonstrate a leading role...” Phrasing should be more cautious as there were no experimental manipulations proving a causal role of the ANT.
- Line 372 – see major Point 1
- Line 414 – “Hence, ..therefore” omit “therefore”.

- Please, provide bin size in the legend to the occurrence probability plots
- Line 620 "...histogram was normalized by the total number of detected scalp SOs (multiplied by 100)..." was normalization done separately for each patient and electrode site?
- The legend to Suppl Table 4 should provide information what is meant by SO-coupling and spindle coupling and what is set to 100% in these couplings.

Reviewer #2 (Remarks to the Author):

This is a very well-written, and thus easy to follow, manuscript that describes the inter-relationship of human SO and sleep spindles in two thalamic regions and their temporal dynamics with cortical SO and SS. The authors need to be congratulated on having grabbed the uncommon chance of recording from thalamic regions in humans. Their results are solid, their analytic and statistical approaches are robust, and their conclusions are supported by the data and do not contain many speculative perspectives. Thus, this manuscript soundly challenges the prevailing view of cortical-only driven SO generation by providing the first evidence in humans that thalamic SO expression, and SO-spindles coupling, is as important as that of the cortex with notable intrathalamic regional differences. Moreover, the earlier start of spindles in ANT with respect to the SO down-to-up state transition is additional support to the finding of the ANT SO preceding cortical SO and to the importance of thalamic spindle drive to the corresponding neocortical SO-coupling.

However, there are a number of issues that the authors should clarify and/or improve upon.

1. Notwithstanding the data in Supplementary Figure 4, it is surprising that no attempt was made to differentiate between fast and slow spindles as the two subtype have different properties and may play diverse roles in memory consolidation. I suggest doing these analyses to search for potential differences in both ANT and MD data, and compared to frontal and parietal neocortical spindles.
2. Are the results of the ANT SO preceding the cortical SO affected if one compares cortical areas that are directly innervated by the ANT with those that are not?
3. The authors should compare SO, spindles and SO-spindle coupling between patients with bipolar contact in ANT and patients with only one bipolar contact in ANT.
4. The potential effects of pharmaco-resistant seizures on the measured sleep parameters should be discussed.
5. Similarly, the potential modulation of sleep architecture and stages by the anti-seizure medications should be discussed and not simply referred to in Supplementary Tables 7 and 8.

Minor points

6. Fig. 1: the description of the images in panel (a) is not clear. Are these real T1-weighted MRI images with superimposed red and blue areas, or else? Also, there is a grey shadow in panel (c) the meaning of which is not clear and not explained in the legend. Please, label appropriately the X axis of panel (b).
7. The results shown in Supplementary Figure 1 are very important: thus, they should be included in Figure 3.
8. Pg 10 Ln 202: not clear what the meaning of “seamlessly” is in this sentence.
9. Pg 15, Ln 291: please abbreviate “phase-slope index” here.
10. Pg 17 Ln 340: “significance” should be “significant”.

Reviewer #3 (Remarks to the Author):

Schreiner et al. present novel and highly interesting findings on the coordination of oscillatory activity during sleep based on intracranial EEG data recorded in the human thalamus, as well as surface EEG. They show that the anterior thalamus organizes cardinal sleep oscillations in humans which has implications not only for the field of sleep physiology but also for the fields of human cognition and memory more generally.

The main novelty lies in demonstrating that SOs not only originate in the neocortex but can likewise be coordinate by thalamic nuclei, and that two thalamic nuclei that have recently been associated with functions of higher cognition and memory (the ANT and MD) may play a key, but differential, roles in coordinating sleep oscillations in the human brain.

The reported dataset is highly valuable and the findings significantly advance our knowledge on how oscillatory activity is organized in the sleeping brain. They also elucidate the functional diversity of specific thalamic nuclei, and propose a critical role of the ANT in memory functions. All analysis are done rigorously, all statistical tests fully support the statements made in the paper. I strongly recommend publication of these results and only have a few minor remarks that the authors could consider to further strengthen their report.

1. The finding that only SOs in ANT (but not in MD) lead neocortical SOs, whereas thalamic sleep spindles preceded their neocortical counterparts in both thalamic nuclei are interesting: Could these differences indicate a functional diversity not only of the thalamic nuclei but also of the sleep oscillations originating there?

2. How can the finding that sleep spindles lock to the same very early phases of SOs in both thalamic nuclei and precede neocortical counterparts be reconciled with the differential orchestrations of SOs between ANT and neocortex vs. MD and neocortex? Are different anatomical loops involved? Fewer/more events coupled? Does this tell us something about the functional diversity of these sleep oscillations?

3. II.252ff The logic of this statement and how it follows from the results reported so far was unclear to me. Do the authors mean that SOs leading in thalamus ANT/neocortex and SOs leading in neocortex for MD/neocortex indicates that both the neocortex and the thalamus can coordinate/initiate sleep oscillations? Could they please elaborate/clarify?

4. II. 381ff The data reported here show that both ANT SOs and spindles lead those in the neocortex. SO-spindle interactions are deemed central to offline memory processing and plasticity during sleep. The ANT has extensive connections to neocortical areas which have previously been perceived to a) be the origin of neocortical SOs and b) are critically involved in memory functions. The data reported in this paper suggest that the SOs detected in central mnemonic processing hubs of the neocortex may instead originate in the thalamus which may have strong implications for its role in coordinating memory reprocessing and storage in the human brain. This is a central point of discussion and could be phrased even more strongly, in my opinion. The results clearly encourage further investigation of a critical role of the thalamus, and ANT in particular, in memory functions.

5. II. 410 and following paragraph: The reasoning why timing of sleep spindles in the SO downstate in thalamus vs. down-to-up-state/up-state in neocortex indicates a concomitant grouping of SOs and spindles by the ANT was not immediately accessible to me (see also I. 421, II.425ff). Why is timing not conserved? Why is timing different in ANT-neo vs. MD-neo? How can this be reconciled? What implications may this have for functional divergence of MD and ANT in the coordination of sleep oscillations?

Could the authors imagine adding a figure that illustrates the interactions they propose to underlie these differential oscillatory dynamics, and how coordination by ANT follows from the observed differences in timing of oscillatory interactions? If these is beyond scope, could the authors add additional information on why the observed timing of oscillations is a necessary pre-requisite to allow for concomitant coordination of neocortical SOs and sleep spindles by ANT?

The following parts would, in my view, benefit from further clarification:

- II. 414: "Hence, the characteristics of SO-spindle interactions in the ANT might therefore highlight how the thalamus concomitantly groups SOs and spindles in the neocortex, leading to the well-known oscillatory nesting."
- II. 421ff "Notably, this divergent property of SO-spindle coordination is a

- 422 necessary pre-requisite to allow for a concomitant coordination of neocortical SOs and spindles by the ANT. Specifically, while we show that both ANT SOs and spindles preceded their neocortical counterparts, the time lag between ANT and neocortical spindles exceeded the time lag between ANT and neocortical SOs. Hence, to enable the prototypical SO-spindle modulation in neocortical circuits (i.e., nesting of spindles towards the SO upstate) under the assumption that both graph-elements in the neocortex are governed by the ANT, it is inevitable that spindles start at relatively earlier phases of the SO in the ANT as compared to spindles of neocortical SO-spindle complexes.”
- II 433ff “Again, in order to project spindles to the neocortex that nest in neocortical up states in a timely manner, a shifted coordination of SOs and spindles in the MD is necessary.”
- II. 438ff “while the interplay of local ANT SO-spindles was tuned in a way that allows for neocortical spindles to nest within SO up states. Importantly, this exact coordination is thought to be instrumental for the consolidation of memories.”

6. II. 414 Stylistic comment: “Hence..., ...therefore...” one of the adverbs could be removed.

Additional questions:

7. Oscillation detection: Does sleep spindle detection work equally well in scalp and thalamic recordings? Have the same thresholds been applied/were approx. the same numbers of events detected? Are slopes and amplitudes comparable? Could ease of detection systematically influence the estimated timing of spindle and SO phase?

8. Is the finding that thalamic spindles (in both ANT and MD) seamlessly transition from the “traditional” frequency range to higher frequencies an issue of power/filtering of spindle oscillations by the skull or a result from different network properties/functional diversity of brain regions?

REVIEWER COMMENTS

Reviewer #1 (Remarks to the Author):

Comment 1: The most important result of the study is that, contrary to the widely held view of a cortical origin of SOs, SOs recorded from ANT precede SOs recorded from frontal/frontotemporal EEG electrodes by on average about 50 ms (as judged from Fig. 2), i.e., a temporal relationship suggesting that ANT SOs drive frontocortical SOs. The lead of ANT SOs is small but, physiologically plausible and highly relevant as to possible cause-effect relationships. The more so the authors are in need to convincingly exclude non-physiological/technical confounds. First, scalp EEG recordings differ from intracranial EEG recordings in that the signal picked up by the electrode, is filtered by skull, tissues etc. which altogether act as low pass filter that may well delay the measured SO negative peak amplitude in the EEG by 50 ms or so, with reference to intracranial recording. Second, the authors argue (line 508) that in order "... to mitigate effects of volume conduction on our analyses, the iEEG recordings were then re-referenced to their immediate neighboring contact (bipolar montage)..." However, the bipolar montage does not entirely exclude that ANT SOs partially represent volume conducted SOs invading ANT from other sites, including anterior neocortex. (To obtain clues as to a possible role of volume conduction, the authors might check for the occurrence of polarity reversals in the SO signal in recordings using a common reference for EEG and iEEG. Events with polarity reversal are more likely reflecting neocortical SOs, and might then be used to estimate potential technical delays in SO peaks between EEG and bipolar iEEG recordings). In sum, these possible technical confounds need to be carefully and convincingly addressed in the Discussion.

Response: We want to thank the reviewer for raising these important points. To examine whether the skull, acting as a low pass filter, might have delayed the signal recorded at the scalp relative to intracranial recordings, we now include a new, independent data set that comprises intracranial recordings from the frontal cortex and simultaneously recorded frontal scalp EEG in three patients. We are confident that we can address the reviewer's point in the most direct way by adding this new data set.

Specifically, we assessed SO coupling between frontal scalp electrodes (electrode FP1) and frontal intracranial contacts in three pre-surgical epilepsy patients (frontal iEEG contacts, $n = 13$) during NREM sleep. Our reasoning was that SOs recorded on FP1 and in intracranial frontal contacts would very likely pick up signals from the same cortical sources, given that iEEG and scalp electrodes are in close proximity. If the skull indeed delays the measured SO negative peak amplitude in the scalp EEG, then we should be able to observe a comparable delay of ~ 50 ms with regards to SOs in these recordings. In contrast, if intracranial and scalp SOs would peak in synchrony, this would indicate that the skull does not delay propagation of SOs and would make a strong point that the temporal difference we observed between ANT and scalp SOs is not primarily driven by the skull acting as a low-pass filter.

Analyses steps used for the new intracranial data were identical to our previous analyses of ANT recordings (i.e., sleep scoring, pre-processing, artifact rejection, SO-detection). On average we detected 1450 ± 137.58 SOs in frontal intracranial contacts during NREM sleep and 1245.7 ± 161.97 SOs in FP1, with a coupling rate of 30.15 ± 1.22 % (these numbers approximately match the numbers described in our ANT data set). As evident in the figure below (a), occurrence probabilities of intracranial detected SO down state peaks relative to scalp detected down state peaks (bin size = 50 ms) indicated that SOs in frontal intracranial and scalp recordings emerged synchronously (positive cluster from -0.1 to 0.1 sec; $p < 0.0009$; time of peak = 0 sec), with no apparent delay.

We then determined the phase of intracranial SOs for all paired SO-events (i.e., all ANT SOs within ± 750 ms of neocortical SOs) at the time of neocortical SO down state. We found a significant nonuniform distribution across contacts (Rayleigh $z = 12.82$, $p < 0.0001$), with the phase of intracranial SOs being

almost identical to the phase of scalp detected events (the phase of neocortical SO down states corresponds to $\pm \pi$; mean coupling direction: $-179.37 \pm 1.84^\circ$; see b).

Based on these new results we are convinced that the observed delay between SOs recorded in ANT and on the scalp is not driven by the low-pass filtering properties of the skull. We added the analyses and results to the Supplementary Information (Supplementary Figure 3) and refer to it in text on page 20, line 413.

Supplementary Fig. 3 | SO coupling frontal iEEG – frontal scalp: To examine whether the skull, acting as a low pass filter, might have delayed the picked-up signal at the scalp level relative to thalamic recordings, an additional, independent data-set was analyzed. Specifically, SO coupling between frontal scalp electrodes (electrode FP1) and frontal intracranial contacts in three pre-surgical epilepsy patients (frontal iEEG contacts, $n = 13$) during NREM sleep was assessed. If the skull indeed delays the measured SO negative peak amplitude at the scalp level then a comparable delay of ~ 50 ms with regards to SOs should be observable in this data, were intracranial and scalp electrodes are in close distance. On average we detected 1450 ± 137.58 SOs in frontal intracranial contacts and 1245.7 ± 161.97 SOs in frontal scalp EEG recordings during NREM sleep, with a coupling rate of $30.15 \pm 1.22\%$. (a) Occurrence probabilities of intracranial detected SO down state peaks relative to scalp detected down state peaks (bin size = 50 ms), indicated that SOs in frontal intracranial and scalp recordings emerged synchronously (positive cluster from -0.1 to 0.1 sec; $p < 0.0009$; time of peak = 0 sec). (b) We then determined the phase of intracranial SOs for all paired SO-events (i.e., all ANT SOs within ± 750 ms of neocortical SOs) at the time of neocortical SO down state. We found a significant nonuniform distribution across contacts (Rayleigh $z = 12.82$, $p < 0.0001$), with the phase of intracranial SOs being almost identical to the phase of scalp detected events (the phase of neocortical SO down states corresponds to $\pm \pi$; mean coupling direction: $-179.37 \pm 1.84^\circ$). (c) Example of NREM sleep segment (15 sec), comprising SOs (top row: scalp recording; bottom row: intracranial). (d) Grand average EEG trace of neocortical SOs (mean \pm SEM, negative peak, time 0). (e) Grand average iEEG trace of intracranial SOs (mean \pm SEM, negative peak, time 0).

We added a paragraph to the discussion (page 20, line 402):

... Since scalp EEG recordings provide an indirect measure of the underlying cortical sources, one might ask whether a delay in cortical SOs relative to ANT SOs might be driven by effects of interjacent tissue (e.g., the skull) delaying the timing of the signal picked up at frontal scalp EEG channels. To mitigate these concerns, we show, in an independent data set, that intracranially recorded SOs in the frontal lobe exhibit no temporal delay with regards to SOs simultaneously recorded by frontal scalp EEG (see Supplementary Fig. 3)...

We also added a paragraph to the methods section (page 26, line 609):

...To investigate whether the signal picked-up at the scalp level might have been delayed by interjacent tissue (e.g., the skull), we assessed SO coupling between frontal scalp electrodes (electrode FP1) and frontal intracranial contacts in an independent data set consisting of three pre-surgical epilepsy patients (for details see Supplementary Fig. 3)...

We also agree with the second point raised by the reviewer: the use of bipolar montage does not entirely exclude that ANT SOs partially represent volume conducted SOs which invaded the ANT from other sites. While the bipolar montage is considered the gold standard in order to resolve local activity¹⁻³, there might still be the chance that volume conducted SOs could intrude the measured signal. To estimate the possible impact of volume conduction on our results we ran a set of control analyses on the data of a sample patient. First, we identified events where SOs are present in the scalp recordings (average referenced as in the main analyses) but not in the bipolar referenced ANT recordings (for an example event see Figure (a) below; for ERPs of all identified events see (c)). Second, we isolated the same events using a common average reference that included both scalp and thalamic recordings (b & d). Our rationale was that SOs which can only be identified in the thalamus after these two steps and that show a of polarity reversal are most likely a product of volume conduction (see b & d). As suggested by the reviewer, we used this approach to estimate potential technical delays in SO peaks between EEG and bipolar iEEG recordings.

Importantly, as illustrated in the Figure below, no delay apart from the polarity reversal was apparent. The insets in (d) show the phase of volume conducted SOs in ANT contacts (common average reference) at the time of neocortical SO downstate in blue, illustrating that the volume conducted SO peaked at almost exactly the opposite phase (Rayleigh $z = 282.71$, $p < 0.00001$; mean direction: 1.59°) as neocortical SOs (Rayleigh $z = 282.71$; $p < 0.00001$; mean direction: 179.45°). Cortical SO down-states using a common average reference (including iEEG as in b and d) peaked at 179.45° (grey inset). We believe that these additional analyses make a strong case against volume conduction as a main driver our main results with regards to SOs.

Supplementary Fig. 4 | Estimating the potential influence of volume conducted SOs on the timing of thalamo-cortical SO coupling: While the bipolar montage represents the gold standard in order to resolve localized activity in intracranial recordings¹⁻³, volume conducted SOs might still intrude the measured signal. To accurately estimate the possible impact of volume conduction on our SO related results we ran additional control analyses on the data of a sample patient. First, we identified events where SOs are present in scalp EEG recordings (average referenced

including scalp EEG electrodes only, as in the main analyses) but not in the bipolar referenced ANT recordings (for an example event see (a); for ERPs of all identified events (scalp in grey, ANT in blue) see (c)). The same events were then isolated using a common average reference that included both scalp EEG and ANT recordings. The rationale of this procedure is to isolate SOs in the ANT that were exclusively a product of volume conduction and by that allowing to estimate potential technical delays in SO peaks between EEG and bipolar iEEG recordings. In sum, no such delay became apparent beyond 180 degree phase shift due to the polarity reversal of the ANT signal in the common average montage. (b) Shows the same example event as in (a) using a common average reference (including both scalp and intracranial recordings). (d) Illustrates all identified events used in (c) again using a common average reference (including both scalp (grey) and intracranial recordings (blue)), highlighting that SOs in scalp recordings and volume conducted SOs in the ANT, exhibiting a reversed polarity, peaked at the same time. The insets show in blue the phase of volume conducted SOs in ANT contacts at the time of neocortical SO downstate, illustrating that volume conducted SOs peaked at almost exactly the opposite phase as neocortical SOs (Rayleigh $z = 282.71$, $p < 0.00001$; mean coupling direction: 1.59°). Cortical SO down-states using a common average reference (including iEEG as in b and d) peaked at 179.45° (Rayleigh $z = 282.71$; $p < 0.00001$).

The following figure shows that the sample patient displayed in Supplementary Fig. 4 above is representative of the main effects with respect to the delay between SOs in ANT and cortex:

Occurrence probabilities of ANT SO down state peaks relative to neocortical SO down state peaks in the sample patient (as in Supplementary Fig. 4), showing a delayed cortical SOs relative to ANT SOs (time = 0; dashed line; bin size = 50 ms), comparable to our main finding (Figure 2a in the main text).

We added a paragraph to the discussion (page. 20, line 408):

...Another concern might arise considering volume conduction contaminating the thalamic signal. We used a bipolar montage for thalamic contacts to ensure a local origin of the signal³. However, there might still be a possibility that the signal recorded from these contacts is contaminated by volume conducted activity from adjacent brain tissue outside the thalamus. We address this issue in a dedicated control analysis (see Supplementary Figure 4) and conclude that none of these factors is likely to systematically bias our findings...

We furthermore added the analyses and results to the Supplementary Information (Supplementary Figure 4) and refer to the new figure in the methods section (page 26, line 596):

...To mitigate effects of volume conduction on our analyses⁴, the iEEG recordings were then re-referenced to their immediate neighboring contact (bipolar montage; see Supplementary Fig. 4 for an estimation of the potential influence of remaining volume conducted SOs on the main results)...

In sum, we want to thank the reviewer again for raising these issues, as we truly believe that the additional analyses presented above do make our main results and the claims of our manuscript stronger.

Comment 2: Although spindles are generally of more local nature than SOs, basically similar questions arise for spindles detected in ANT and MD: They may represent spindle originating from the respective thalamic region but, volume conduction cannot be excluded. In this regard, separate analyses of spindles that do and do not cooccur in phase in ANT and MD recordings (the later representing localized thalamic spindles) might provide further insights into a possible “guiding” role of ANT in organizing the SO-spindle interplay.

Response: This is a good point. To test this notion, we isolated spindles in the ANT whose peak (maximally negative amplitude) overlapped with MD spindle peaks ± 50 ms (‘local’ spindles 267.91 ± 51.57 ANT spindles out of 2203.8 ± 213.5 ; $11.78 \pm 1.84\%$). Then we conducted separate analyses for those ANT spindles that overlapped with MD spindles (non-local spindles) and those that occurred in isolation (local spindles). As shown in the figure below, the same temporal dynamics with regards to thalamo-cortical spindle coupling and SO-spindle coupling became apparent for both local and non-local ANT spindles. Specifically, in both cases (local and non-local spindles) thalamic spindles precede neocortical spindles and preferentially emerge around ANT SO down-states. This lack of difference between local ANT sleep spindles in their temporal dynamics is line with our general results that thalamo-cortical spindle dynamics and SO-spindle dynamics were quite comparable between the ANT and the MD. As outlined in the discussion (page 22, line 458) both the ANT and the MD are innervated by the nucleus reticularis (TRN), with burst firing in circuits between thalamo-cortical (e.g., ANT & MD) and thalamic reticular neurons generating sleep spindles⁵. Hence, the basic mechanisms of sleep spindle generation are expected to be highly similar across different thalamic nuclei.

Supplementary Fig. 1 | Local and non-local ANT spindles. Non-local ANT spindles were defined as spindles identified in the ANT whose peak (maximally negative amplitude) overlapped with MD spindle peaks ± 50 ms (267.91 ± 51.57 ANT spindles out of 2203.8 ± 213.5 ; $11.78 \pm 1.84\%$). All other spindles identified in the ANT were defined as local ANT spindles. The same temporal dynamics with regards to thalamo-cortical spindle coupling and SO-spindle coupling were found for local and non-local ANT spindles. **(a)** Occurrence probabilities of local ANT spindle peaks relative to neocortical spindle peaks (maximal negative amplitude, time = 0; dashed line; bin size = 50 ms), indicating that local ANT spindles precede neocortical spindles (positive clusters from -0.3 to -0.05 sec, $p < 0.0001$ & 0.2 to 0.25 sec, $p = 0.023$; time of peak: -0.1 sec). **(b)** Local ANT spindles preferential emerged around ANT SO down states (-0.15 to 0.25 sec; $p = 0.009$; peak: 0 sec). **(c)** Phases of the SO-spindle modulation derived from local ANT events. Local ANT spindles started specifically around the ANT SO down states (mean coupling direction: $-168.93^\circ \pm 10.10^\circ$; Rayleigh $z = 7.93$, $p < 0.0001$). **(d)** Occurrence probabilities of non-local ANT spindle peaks relative to neocortical spindle peaks, indicating that non-local ANT spindles likewise precede neocortical spindles (positive cluster from -0.15 to 0 sec, $p < 0.017$; time of peak: -0.05 sec). **(e)** Non-local ANT spindles also preferential emerged around ANT SO down states (-0.05 to 0.1 sec; $p = 0.017$; peak: -0.05 sec). **(f)** Phases of the

SO-spindle modulation derived from non-local ANT events. Non-local ANT spindles started specifically around the ANT SO down states (mean coupling direction: $-157.91^\circ \pm 10.21^\circ$; Rayleigh $z = 7.86$, $p < 0.0001$).

We ran the same analysis as above for ‘local’ and ‘non-local’ MD spindles (local spindles: 149.83 ± 35.69 MD spindles out of 2127.8 ± 238.2 ; 7.04%). As shown below, also in the case of the MD local and non-local spindles precede neocortical spindles and preferentially emerge around MD SO down-states.

Supplementary Fig. 2 | Local and non-local MD spindles. Non-local MD spindles were defined as spindles identified in the MD whose peak (maximally negative amplitude) overlapped with ANT spindle peaks ± 50 ms (149.83 ± 35.69 MD spindles out of 2127.8 ± 238.2 ; 7.04%). All other spindles identified in the ANT were defined as local ANT spindles. The same temporal dynamics with regards to thalamo-cortical spindle coupling and SO-spindle coupling were found for local and non-local MD spindles. **(a)** Occurrence probabilities of local MD spindle peaks relative to neocortical spindle peaks (maximal negative amplitude, time = 0; dashed line; bin size = 50 ms), indicating that local MD spindles precede neocortical spindles (positive cluster from -0.25 to 0 sec, $p < 0.0001$; time of peak: -0.05 sec). **(b)** Local ANT spindles preferential emerged around ANT SO down states (-0.25 to 0.05 sec; $p < 0.0001$; peak: -0.05 sec). **(c)** Phases of the SO-spindle modulation derived from local ANT events. Local ANT spindles started specifically around the ANT SO down states (mean coupling direction: $-172.05^\circ \pm 7.60^\circ$; Rayleigh $z = 9.59$, $p < 0.0001$). **(d)** Occurrence probabilities of non-local ANT spindle peaks relative to neocortical spindle peaks, indicating that non-local ANT spindles likewise precede neocortical spindles (positive cluster from -0.1 to 0 sec, $p < 0.0001$; time of peak: -0.05 sec). **(e)** Non-local ANT spindles also preferential emerged around ANT SO down states (-0.2 to 0 sec; $p = 0.017$; peak: -0.3 sec). **(f)** Phases of the SO-spindle modulation derived from non-local ANT events. Non-local ANT spindles started specifically around the ANT SO down states (mean coupling direction: $-155.38^\circ \pm 8.68^\circ$; Rayleigh $z = 8.91$, $p < 0.0001$).

We added the analyses and results to the Supplementary Information (Supplementary Figure 1 + 2). We refer to this new information in the results section (page 10, line 220):

... see Supplementary Fig. 1 for results concerning spindles detected simultaneously in ANT and MD and spindles detected in ANT only)...

and page 12, line 261:

...Supplementary Fig. 2 for results concerning spindles detected simultaneously in MD and ANT and spindles detected in MD only...

We also mention this in the Discussion, page. 22, line 466:

...In line with previous work on spindles in posterior thalamic nuclei⁶, we identify local and non-local spindles in ANT and MD. With respect to the neural circuitry generating spindles in ANT and MD, one could assume recurrent loops between TRN and thalamocortical neurons, similar to what has been shown in animals^{7,8}. Extrapolating these findings from animals to humans needs to be done with caution, however, since result from animal models concerning TRN innervation of thalamic nuclei is not always univocal⁶. In general, it is assumed that TRN is organized in sectors that topographically project to thalamic nuclei¹⁰, which can explain why local spindles emerge in both MD and ANT. However, the mapping of TRN sections to thalamic nuclei is not one-to-one¹¹: for example, some TRN sectors projecting to the MD also project to ANT¹²⁻¹⁴. Such overlapping projections seem plausible when considering that MD and ANT thalamocortical neurons share the prefrontal cortex as a common target...

Comment 3: As judged from Fig 6 and 7, spindles recorded in ANT and MD appear to cover the downstate of the SOs recorded in the same region. Assuming that SO down-states represent synchronized membrane hyperpolarizations, it is difficult to understand that SOs and spindles reflect potential activity in the same cells circuits. The excitable troughs of the spindle oscillations are typically accompanied by bursts of spikes considered to be driven by the depolarizing phase of the SO upstate. Is there a plausible mechanism that would explain the occurrence of a spindle in a strongly hyperpolarized network?

Response:

We want to thank the reviewer for this important question. Sleep spindles emerge when a certain hyperpolarization level is achieved in the thalamus¹⁵. Specifically, thalamic hyperpolarization deactivates I_T and activates I_h channels, enabling rhythmic burst firing and hence giving rise to thalamic spindles⁵. It is generally assumed that during NREM sleep, the thalamus is in a sufficient hyperpolarized state that spindles emerge spontaneously⁵, hence regardless of the presence of SO down-states. Opposing this view, Mak-McCully and colleagues¹⁶ have shown that thalamic down-states in humans are tightly coupled to the onset of sleep spindles (both recorded from the pulvinar), indicating that human thalamo-cortical neurons might require additional hyperpolarization by down-states to trigger the spindle generating cascade. Our results are very much in line with the findings reported by Mak-McCully and colleagues, in showing that thalamic sleep spindle onsets overlap with the presence of SO down-states (see below for a comparison of results; (a) SO-spindle coupling in the ANT in our data; (b) SO-spindle coupling in three example patients in the pulvinar copied from Mak-McCully et al., 2017¹⁶, Figure 2; left column in (b) depicts spindle onsets).

We now point this out in the discussion, page 23, line 518:

...The tight coupling of thalamic spindles to thalamic down-states is in line with previous work showing a very similar temporal relationship in the human pulvinar⁴⁶. These results indicate that the hyperpolarization during thalamic down-states is required to trigger the neuronal cascade underlying spindle generation...

Minor points:

- Abstract/Results/Discussion should provide average durations of the time lags between SO/spindles between thalamic and EEG recordings.

Response: We now provide the average durations of the time lags throughout the manuscript.

- Line 43, studies of thalamic SOs are not so “recent”.

Response: We removed the word recent.

- Please, add reference to the table of patient data in the beginning of Results section

Response: We referenced all Supplementary tables, including the table about patient data, in the first results section (page 5, lines 78 - 90)...

*To examine thalamo-cortical interactions during human NREM sleep, we analyzed simultaneously-recorded intracranial thalamic and scalp EEG collected across 11 full nights of sleep from 8 patients with pharmaco-resistant epilepsy. On average, participants slept for 8.82 ± 1.41 hours, with 63.51 ± 0.07 % spent in NREM sleep (stages N2 and N3; **see Supplementary Table 1** for proportions of times spent in each stage, Fig. 1c). Thalamic activity was recorded from the ANT and the MD (see Fig. 1a and methods for details on electrode localization). We isolated sleep oscillations from 27 bipolar intracranial thalamic channels (12 ANT; 15 MD) and scalp EEG electrodes (**see Supplementary Table 2** for details). To determine thalamo-cortical coordination during NREM sleep and their specificity, we detected SOs, spindles and SO-spindle events independently in thalamic iEEG (ANT & MD) and scalp EEG recordings, assessed their coupling and delineated the temporal relationship of thalamo-cortical interactions (for descriptive values and characteristics **see Supplementary Tables 3-6**; Fig. 1b).*

- Line 120 “...time locked to SO down states”, should be phrased more precisely “...time locked to the SO down state peaks”

Response: We changed the wording accordingly.

- Reporting results of EEG SOs and spindles the authors should add where the recordings were performed (F/FT for SOs, P for spindles).

Response: We added the relevant information at the beginning of each results section.

- The vertical zero line in the TF plots (e.g., Fig. 3e) is difficult to see

Response: We intensified the vertical zero line in each figure to facilitate visibility.

- Line 189 “...clusters: -2.5 to -0.05 s” 0.25?

Response: Thank you for spotting the mistake. The correct cluster time is -0.25 to 0.05. We changed the text accordingly.

- Fig 4a, there is a second significant peak on the positive side not mentioned in the text.

Response: We refer to the second peak both in the main text and the figure legend

Main text (page 10, lines 197-200):

... Here, we show that the human ANT is putatively part of the spindle generating thalamic circuit, as ANT spindle occurrence peaked on average before the emergence of neocortical spindles (Fig. 4a; positive clusters: -0.25 to 0.05 s, $p < 0.005$ & 0.15 to 0.2 s, $p = 0.019$; corrected for multiple comparisons across time; time of peak: -0.1 sec) ...

Figure legend:

... Fig 4. | ANT spindles lead neocortical spindles. (a) Occurrence probabilities of ANT spindle peaks relative to neocortical spindle peaks (maximal negative amplitude, time = 0; dashed line; bin size = 50 ms), indicating that ANT spindles precede neocortical spindles. The solid black line indicates significant differences, resulting from comparison with spindle-free control events (positive clusters from -0.25 to -0.05 sec, $p < 0.005$ & 0.15 to 0.2 sec, $p = 0.019$; corrected for multiple comparisons across time; time of peak: -0.1 sec)

- Figure 4d shows grand average for spindles. Please indicate the n for these averages. I am surprised to see that several cycles of the spindle oscillations are preserved in this grand average although the frequency of individual spindles probably varies (to a certain degree) within and across the patients, such that the discrete oscillations are typically smeared in grand averages.

Response: We now added the mean number of events (\pm SEM) to all ERP related figure legends (please note that these descriptive values are also specified in Supplementary Tables 3-6).

It is correct that the individual peak spindle frequency varies across patients (please see Supplementary Table 6 for details), but this variance is rather small (13.17 ± 0.09 for ANT, range [12 to 14 Hz]; 13.03 ± 0.11 for MD, range [12 to 14 Hz]). Please note that Fig. 4d depicts spindle-peak locked averages. Hence, large deflections at the peak of individual spindles will be preserved in the grand average, and small variations in frequency will cancel each other out after more strongly after multiple cycles of the spindle. To highlight this, we plot a single subject average next to the group level grand average (main Figure 4f) below.

Very similar effects can be found in independent datasets acquired in epileptic patients (pasted from Figure 1 in Helfrich et al., 2019¹⁷).

- single subject -
scalp

- group level -
scalp

ANT

ANT

[note that (a) was high-pass filtered at 3 Hz, which eliminates the slow wave component visible in (b)]

- Spindle-SO coupling was performed with reference to the spindle onset. I didn't find a definition of the spindle onset. Please, add.

Response: Thank you for spotting this. We added the respective information to the methods section (pages 28, line 664):

... The root mean square (RMS) signal was calculated for the filtered signal at each channel using a moving average of 200 ms, and spindles that exceeded the mean amplitude of all detected events by 1.5 SD entered the analysis. Whenever the signal exceeded this threshold for more than 0.5 sec but less than 3 sec (duration criteria), a spindle event was detected. Spindle onsets were defined as the points in time when the signal initially crossed the threshold. Epochs, time-locked to the maximally negative spindle trough (-2.5 to +2.5 sec), were extracted from the unfiltered raw signal for subsequent time-locked analyses...

- Line 313 "...power peaks of MD sleep spindles started around MD.." I can't see this in Fig 7 where power is increased distinctly before.

Response: The reviewer is correct. We rephrased the sentence accordingly (page 16, line 321).

... MD spindles, however, started preferentially briefly before the MD SO down states (-0.35 to 0 s; $p < 0.0001$; peak: -0.05 s; Fig 7b) ...

- Line 344 "Our results demonstrate a leading role..." Phrasing should be more cautious as there were no experimental manipulations proving a causal role of the ANT.

Response: We toned the starting sentence of the discussion down. The sentence now reads as follows (line 364):

...Our results suggest that anterior thalamic activity might play a fundamental role in the orchestration of cardinal sleep rhythms in humans...

- Line 372 – see major Point 1

Response: We address the topic concerning the bipolar montage now specifically in the discussion (see response to Comment 1).

- Line 414 – "Hence, ..therefore" omit "therefore".

Response: We removed the word "therefore".

- Please, provide bin size in the legend to the occurrence probability plots

Response: We add the bin size (= 50 ms) to every figure legend in the main text and the Supplementary Information.

- Line 620 "...histogram was normalized by the total number of detected scalp SOs (multiplied by 100)..." was normalization done separately for each patient and electrode site?

Response: Yes, normalization was always done separately per participant and electrode site. We added this information (page 29, line 720):

... The resulting histogram was normalized by the total number of detected scalp SOs (multiplied by 100) per participant and electrode site...

- The legend to Suppl Table 4 should provide information what is meant by SO-coupling and spindle coupling and what is set to 100% in these couplings.

Response: SO- and spindle coupling in Supplementary tables 3 and 4 refers to the percentage of thalamic SOs and spindles coupled to scalp detected SOs / spindles and vice versa. The overall number of SOs and spindles at a given recording site (i.e., ANT, MD & scalp) was set to 100 percent.

The table legends now read as follows (pages 38 and 39):

Supplementary Table 3 | Sleep Oscillations ANT: Data are means \pm s.e.m. Number of detected events and percentage of paired events (i.e., percentage of co-occurring events across sites, relative to the overall number of events at given site) in NREM sleep at ANT contacts and scalp electrodes. Density (events/min of NREM sleep).

Supplementary Table 4 | Sleep Oscillations MD: Data are means \pm s.e.m. Number of detected events and percentage of paired events (i.e., percentage of co-occurring events across sites, relative to the overall number of events at given site) in NREM sleep at MD contacts and scalp electrodes. Density (events/min of NREM sleep).

Reviewer #2 (Remarks to the Author):

Comment 1: Notwithstanding the data in Supplementary Figure 4, it is surprising that no attempt was made to differentiate between fast and slow spindles as the two subtype have different properties and may play diverse roles in memory consolidation. I suggest doing these analyses to search for potential differences in both ANT and MD data, and compared to frontal and parietal neocortical spindles.

Response: We want to thank the reviewer for this thoughtful comment. As suggested by the reviewer, we now divided the sleep spindle band into slow and fast spindles (9-12 Hz and 12-16 Hz, respectively^{18,19}). We then assessed the thalamo-cortical coupling specifically for slow and fast spindles with respect to the scalp electrodes used in our main analyses (see Supplementary table 2 for details) as well as frontal and parietal sites.

Results with regards to fast spindle coupling between the thalamus (computed separately for ANT and MD) and scalp EEG (scalp electrodes used in the main analysis, frontal and parietal sites) were highly comparably to our original spindle results. Occurrence of both ANT and MD spindles peaked on average before the emergence of cortical spindles (irrespective of site; see below as well as Supplementary Fig. 10 and 11 (a, d + g)). Results with regards to slow spindles were less stable. While the same distribution, with slow thalamic spindles leading cortical spindles became apparent descriptively, the outcomes were of modest stability when tested against event-free occurrence probabilities (see below as well as Supplementary Fig. 10 and 11 (b, e + h)). When directly comparing ANT-neocortical occurrence probabilities for fast and slow spindles, no significant difference was found (see Supplementary Fig. 10 (c, f + i)). In contrast, when comparing MD-neocortical occurrence probabilities for fast and slow spindles, several negative differences were significant. All of the significant differences were outside the relevant time-range of spindle coupling (i.e., peak \pm 500ms; see Supplementary Fig. 11 (c, f + i)). The differences around time=0 did not reach significance in the cluster-based permutation approach (all p-values $>.08$).

In addition, we added descriptive values (spindle numbers, densities, coupling rate and peak frequencies) with regards to slow and fast spindles to the Supplementary (Supplementary table 9) and refer to the newly added information in text.

The reference in text reads as follows (page 29, line 701):

...(ANT or MD, respectively; see Supplementary Table 2 for an overview of the selected scalp electrodes; see Supplementary Fig. 9 for spindle-spindle occurrence probabilities, where scalp spindles were exclusively derived from frontal or parietal electrodes; for slow- and fast- spindle related results see Supplementary Fig. 10, 11 and Supplementary table 9)...

Supplementary Fig. 10 | ANT - cortical coupling for fast and slow spindles: Occurrence probabilities of ANT fast (a) and slow spindle (b) peaks relative to cortical spindles (same electrode sites as in the main analyses; for details see Supplementary table 2; maximal negative amplitude, time = 0; bin size = 50 ms). Both fast and slow ANT spindles significantly precede neocortical fast and slow spindles (fast spindles: first positive cluster from -0.3 to -0.1 sec, $p = 0.0007$; second cluster from 0.15 to 0.2 sec, $p = 0.015$; time of peak: -0.1 sec; slow spindles: positive cluster from -0.1 to 0.05 sec, $p = 0.001$). (c) the comparison of fast ANT-neocortical (dark blue) and slow ANT-neocortical SO interactions (light blue), did not yield significant differences ($p > 0.05$). (d + e) Occurrence probabilities for ANT fast and slow sleep spindles with regards to frontal neocortical fast and slow spindles. Fast ANT spindles significantly precede neocortical frontal fast spindles (first positive cluster from -0.4 to -0.05 sec, $p < 0.0001$; second cluster from 0.15 to 0.2 sec, $p = 0.008$; time of peak: -0.1 sec). In case of slow spindles, no significant differences were observable when tested against event-free occurrence probabilities ($p = 0.06$). (f) the comparison of frontal fast ANT-neocortical (dark blue) and slow ANT-neocortical SO interactions (light blue), did not yield significant differences ($p > 0.05$) (g + h) Occurrence probabilities for ANT fast and slow sleep spindles with regards to parietal neocortical fast and slow spindles. Fast ANT spindles significantly precede neocortical parietal fast spindles (first positive cluster from -0.25 to -0.2 sec, $p = 0.008$; second cluster from -0.1 to 0.05 sec, $p = 0.0002$; time of peak: -0.1 sec). In case of slow spindles, no significant differences were observable when tested against event-free occurrence probabilities ($p = 0.09$). (i) the comparison of parietal fast ANT-neocortical (dark blue) and slow ANT-neocortical SO interactions (light blue), did not yield significant differences ($p > 0.05$).

Supplementary Fig. 11 | MD - cortical coupling for fast and slow spindles: Occurrence probabilities of MD fast (a) and slow spindle (b) peaks relative to cortical spindles (same electrode sites as in the main analyses; for details see Supplementary table 2; maximal negative amplitude, time = 0; bin size = 50 ms). Both fast and slow MD spindles significantly precede neocortical fast and slow spindles (fast spindles: positive cluster from -0.3 to 0.2 sec, $p < 0.0001$; slow spindles: positive cluster from -0.1 to -0.05 sec, $p = 0.006$). (c) the comparison of fast MD-neocortical (dark red) and slow MD-neocortical SO interactions (light red), yielded multiple significant differences (negative cluster 1 from -0.75 to -0.5 sec, $p > 0.0001$; negative cluster 2 from -1.1 to -0.95 sec, $p = 0.002$; negative cluster 3 from 1.2 to 1.3 sec, $p = 0.004$). (d + e) Occurrence probabilities for MD fast and slow sleep spindles with regards to frontal neocortical fast and slow spindles. Both fast and slow MD spindles significantly precede neocortical frontal fast and slow spindles (fast spindles: positive cluster from -0.35 to 0.3 sec, $p < 0.0001$, time of peak: -0.05 sec; slow spindles: first positive cluster from -0.25 to -0.15 sec, $p = 0.002$; second positive cluster from -0.05 to 0.15 sec, $p < 0.0001$; time of peak: -0.05). (f) the comparison of fast frontal MD-neocortical (dark red) and slow MD-neocortical SO interactions (light red), yielded two negative clusters (negative cluster 1 from -0.95 to -0.7 sec, $p > 0.0001$; negative cluster 2 from -1.35 to -1.15 sec, $p > 0.0001$). (g + h) Occurrence probabilities for MD fast and slow sleep spindles with regards to parietal neocortical fast and slow spindles. Both fast and slow MD spindles significantly precede neocortical parietal fast and slow spindles (fast spindles: positive cluster from -0.25 to 0.15 sec, $p < 0.0001$, time of peak: -0.1 sec; slow spindles: positive cluster from -0.1 to -0.05 sec, $p = 0.006$; time of peak: -0.05). (i) the comparison of fast parietal MD-neocortical (dark red) and slow MD-neocortical SO interactions (light red), one two negative clusters (negative cluster 1 from -0.7 to -0.5 sec, $p > 0.0001$).

Supplementary Table 9 | Descriptives Slow and Fast spindles: Data are means \pm s.e.m. Number of detected events and percentage of paired events between thalamic contacts and scalp electrodes. Density (events/min).

ANT spindle	slow	fast	t	P
spindle number	1850.5 \pm 132.8	1829.3 \pm 155.1	0.13	0.85
spindle density	7.47 \pm 0.34	7.55 \pm 0.58	-0.1	0.87
spindle-coupling [%]	23.5 \pm 3.1	31.7 \pm 3.9	-1.8	0.08
Peak Freq. Hz	10.29 \pm 0.15	14.02 \pm 0.11		
MD spindle				
spindle number	1990.6 \pm 132.8	1945.1 \pm 222.9	0.32	0.70
spindle density	8.56 \pm 0.97	8.62 \pm 0.57	-0.1	0.92
spindle-coupling [%]	27.0 \pm 2.7	26.4 \pm 5.2	0.011	0.90
Peak Freq. Hz	10.20 \pm 0.12	14.08 \pm 0.12		

Comment 2: Are the results of the ANT SO preceding the cortical SO affected if one compares cortical areas that are directly innervated by the ANT with those that are not?

Response: In the original analysis, we selected frontal scalp electrodes, due to the supposed role of the frontal cortex in the generation of SOs (e.g., ^{20,21}) and its innervation by the ANT ^{22,23}. We now additionally assessed occurrence probabilities of ANT SOs relative to neocortical SOs derived from occipital electrodes (i.e., electrode O2), as none of the main ANT-cortical projections innervates the occipital cortex²⁴ (but see ²⁵). Please note that we had to remove 2 patients (4 ANT contacts) from this analysis due to missing occipital electrodes. We detected occipital SOs that did not overlap with frontal detected SOs (\pm 750 ms, $n = 1054.91 \pm 118.38$) to ensure that the isolated SOs were not mainly products of volume conduction. Accordingly, different ANT SOs were coupled to either occipital or frontal SOs. Interestingly, SOs in the ANT did not precede, but peaked in parallel to occipital derived SOs, as evidenced by occurrence probabilities (see Figure below; positive cluster from -0.05 to 0.1 sec, $p = 0.01$; time of peak: 0.0 sec). We also determined the phase of thalamic SOs for all paired SO-events at the time of occipital SO down state (\pm 750 ms, $n = 176.12 \pm 29.71$; note that only 16.31 ± 1.88 % of ANT SOs were coupled to occipital derived SOs as compared to 33.74 ± 3.02 % in case of frontal SOs; $t = -4.37$, $p = 0.0003$). We found a significant nonuniform distribution across contacts (Rayleigh $z = 7.24$, $p < 0.001$), with the phase of ANT SOs following their occipital counterparts (mean coupling direction: $166.56 \pm 6.20^\circ$). Next, we directly compared these outcomes to the results of our main analyses (main figure 2 a + b). Occurrence probabilities for the comparison of ANT-frontal and ANT-occipital SO interactions differed significantly from -0.25 to -0.05 sec ($p = 0.013$), showing that ANT SOs preferentially emerged earlier with regards to frontal scalp SOs as compared to occipital scalp SOs. Comparing the phase distributions for the coupling between thalamus and neocortex in relation to frontal and occipital SOs yielded a statistical trend in the direction that ANT SOs coupled to earlier phases of frontal SOs as compared to occipital SOs (Watson-Williams test: $F = 4.12$, $p = 0.056$). These outcomes suggest that overall, the association of ANT SOs to occipital SOs is considerably smaller as compared to frontally derived SOs, as observable in significantly smaller coupling rates. This could be due to the frontal cortex being directly innervated by the ANT. In addition, also the outcome that ANT SOs did not precede occipital SOs provides a further hint towards the specificity of ANT – frontal interactions with regards to SOs.

We added the analyses and results to the Supplementary Information (Supplementary Figure 8 and refer to it in text; page 29, line 701).

The reference in text reads as follows:

...(see Supplementary Table 1 for an overview of the selected scalp electrode; see Supplementary Fig. 7 for ANT SO occurrence probabilities relative to scalp electrode Fz; see Supplementary Fig. 8 for ANT SO occurrence probabilities relative to occipital scalp electrodes)...

Supplementary Fig. 8 | SO coupling ANT-Occipital: To assess the specificity of ANT – frontal interaction with regards to SOs (Figure 2 a + b), the occurrence probabilities of ANT SOs relative to neocortical SOs derived from occipital electrodes (i.e., electrode O2) were evaluated. Due to missing / noisy occipital electrodes 2 patients (4 ANT contacts) had to be removed from this analysis. Occipital SOs that did not overlap with frontal detected SOs (± 750 ms, $n = 1054.91 \pm 118.38$) were isolated, to ensure that the captured SOs are not mainly products of volume conduction. **(a)** Occurrence probabilities revealed, that SOs in the ANT did not precede, but peaked in parallel to occipital derived SOs (positive cluster from -0.05 to 0.1 sec, $p = 0.01$; time of peak: 0.0 sec). **(b)** The phase of thalamic SOs for all paired SO-events at the time of occipital SO down state (± 750 ms, $n = 176.12 \pm 29.71$; note that only 16.31 ± 1.88 % of ANT SOs were coupled to occipital derived SOs as compared to 33.74 ± 3.02 % in case of frontal SOs; $t = -4.37$, $p = 0.0003$). A significant nonuniform distribution across contacts (Rayleigh $z = 7.24$, $p < 0.001$) was detectable, with the phase of ANT SOs following their occipital counterparts (mean coupling direction: $166.56 \pm 6.20^\circ$). **(c)** Direct comparison of occurrence probabilities for ANT-occipital and ANT-frontal SO interactions revealed a significant difference between the two distributions from -0.25 to -0.05 sec ($p = 0.013$), indicating that ANT SOs preferentially emerged earlier with regards to frontal scalp SOs (light blue) as compared to occipital scalp SOs (dark blue). **(d)** Comparing the phase distributions for the coupling between thalamus and neocortex in relation to frontal and occipital SOs yielded a statistical trend in the direction that ANT SOs coupled to earlier phases of frontal SOs as compared to occipital SOs (Watson-Williams test: $F = 4.12$, $p = 0.056$).

Comment 3: The authors should compare SO, spindles and SO-spindle coupling between patients with bipolar contact in ANT and patients with only one bipolar contact in ANT.

Response: We thank the reviewer for this interesting comment. As suggested by the reviewer, we now assessed thalamo-cortical dynamics for SOs, spindles and SO-spindle coupling in patients where both

contacts ($n = 8$ pairs, “full ANT”) or just one contact of a bipolar pair ($n = 4$ contact pairs, “single ANT”) were localized to the ANT.

Overall, ANT-scalp dynamics with regards to SOs, spindles and the coupling between SOs and spindles were qualitatively highly similar between both groups (see figure below). As in the main analyses, SOs and spindles detected in the ANT preceded their scalp counterparts and spindles in the ANT were initiated around SO down-states. However, these results were only found to be significant in full ANT comparisons (for details please see below), while none of the comparisons involving single ANT contact pairs reached significance. That said, also none of the direct comparisons between full ANT and single ANT pairs reached significance, indicating that the lack of an effect with regards to single ANT pairs might rather be driven by a lack of power ($n = 4$ pairs). We added the analyses and results to the Supplementary Information (Supplementary Figure 5 and refer to it in text; page 26, line 599).

The reference in text reads as follows:

...A bipolar pair of contacts was considered to be in the ANT if both or at least one contact was localized to the ANT, but no contact was localized in the MD (see Supplementary Fig. 5 for a comparison of full ANT contacts and single ANT contacts with regards to occurrence probabilities of SOs, spindles and SO-spindles)...

Supplementary Fig. 5 | ANT-scalp dynamics with regards to contact pairs were both contacts ($n = 8$; full ANT) or just one contact of a bipolar pair ($n = 4$, single ANT) were localized to the ANT: (a) Occurrence probabilities of full ANT SO down state peaks relative to scalp derived SO down state peaks (bin size = 50 ms), indicated that full ANT SOs preceded scalp derived SOs (positive cluster from -0.15 to 0 sec; $p < 0.014$; time of peak = -0.05 sec). (b) Occurrence probabilities of single ANT SO down state peaks relative to scalp derived SO down state peaks (bin size

= 50 ms). While single ANT SOs preceded descriptively scalp derived SOs, no significant difference was observable when tested against event-free occurrence probabilities ($p > 0.1$). (c) No significant difference became apparent when directly comparing full ANT-scalp occurrence probabilities (green) and single ANT-scalp occurrence probabilities (orange). (d) Occurrence probabilities of full ANT spindle peaks relative to neocortical spindle peaks (maximal negative amplitude, time = 0; dashed line; bin size = 50 ms), indicating that full ANT spindles precede neocortical spindles (positive clusters from -0.35 to -0.2 sec, $p < 0.001$ & -0.1 to 0.05 sec, $p < 0.001$; time of peak: -0.05 sec). (e) Again, single ANT sleep spindles, exhibited descriptively the same dynamics as full ANT spindle, but did not reach significance when tested against event-free occurrence probabilities ($p > 0.1$). (f) No significant difference became apparent when directly comparing spindle related full ANT-scalp occurrence probabilities (green) and single ANT-scalp occurrence probabilities (orange). (g) Occurrence probabilities of full ANT spindle onsets with respect to full ANT SO down states (bin size = 50 ms), illustrating that spindles preferentially emerged around ANT SO down states (-0.05 to 0 sec; $p = 0.048$). (h) The same SO-spindle related dynamic became apparent with regards to single ANT contact pairs, but again did not reach significance when tested against event-free occurrence probabilities ($p > 0.1$). (i) Direct comparison of full and single ANT SO-spindle dynamics (green and orange, respectively) did not yield any significant difference ($p > 0.1$).

Comment 4: + 5 The potential effects of pharmaco-resistant seizures on the measured sleep parameters should be discussed. Similarly, the potential modulation of sleep architecture and stages by the anti-seizure medications should be discussed and not simply referred to in Supplementary Tables 7 and 8.

Response: We want to thank the reviewer for this thoughtful comment. We fully agree that seizures as well as anti-seizure medication might affect sleep parameters. In general, results from medicated epilepsy patients, having just undergone invasive surgery, warrant caution when generalizing to healthy population. That said, our patient population was heterogeneous in terms of medication, age, type of epilepsy and affected cortical areas, indicating that no single particular aspect should have biased our findings. In addition, while only seizure-free nights were included in the analysis (we added this information to the methods section; see page 27, line 635), our stringent artifact rejection procedure makes it unlikely that our results are driven by epileptiform activity.

Bolstering this assumption, sleep architecture (see below and Supplementary table 1) and SO- and spindle numbers and densities (for better comparability across studies we now added event densities to Supplementary Table 3 & 4; also see below) were within normal ranges of healthy participants. Together, these results suggest that neither medication nor epileptic activity might have systematically biased our results. Nevertheless, we now added these import topics to the discussion.

The new section reads as follows (page 20, line 417):

...In addition, our data were recorded in epilepsy patients in whom anti-epileptic medication and epileptiform events may affect sleep architecture and sleep related oscillations²⁶⁻²⁹, warranting caution when generalizing to healthy population. However, our patient population was heterogeneous in terms of medication, age, epilepsy form and affected cortical area. Furthermore, sleep architecture and SO- and spindle quantity and densities were within normal ranges of healthy participants (see Supplementary tables 3 and 4,^{30,31}), making it unlikely that a single aspect had a systematic impact on our results. In addition, while only seizure-free nights were included in the analysis, our strict artifact rejection procedure minimized the impact of epileptiform activity on our findings...

Supplementary Table 1 | Sleep architecture: Data are means \pm s.e.m. N1, N2: NREM sleep stages N1 & N2, SWS: slow-wave sleep, REM: rapid eye movement sleep, WASO: wake after sleep onset. TST: total sleep time (in minutes).

	N1	N2	SWS	REM	WASO	TST [min]
Sleep stage [%]	6.1 \pm 1.7	47.4 \pm 4.2	16.5 \pm 3.5	12.7 \pm 2.2	15.9 \pm 3.5	529.4 \pm 30.1

Supplementary Table 3 | Sleep Oscillations ANT: Data are means \pm s.e.m. Number of detected events and percentage of paired events in NREM sleep at ANT contacts and scalp electrodes. Density (events/min)

	ANT	Scalp	t	P
SO number	1573.8 \pm 129.9	1513.5 \pm 130.1	1.11	0.29
SO density	6.29 \pm 0.21	5.81 \pm 0.24	1.32	0.21
SO-coupling [%]	33.74 \pm 3.02	31.11 \pm 3.25	2.07	0.06
spindle number	2203.8 \pm 213.5	1903.3 \pm 211.5	3.19	0.008
spindle density	8.83 \pm 0.21	6.09 \pm 1.08	2.56	0.017
spindle-coupling [%]	43.8 \pm 4.2	37.3 \pm 4.1	3.43	0.005
SO-spindle number	431 \pm 52.1	304.1 \pm 69.8	3.18	0.008

Supplementary Table 4 | Sleep Oscillations MD: Data are means \pm s.e.m. Number of detected events and percentage of paired events in NREM sleep at MD contacts and scalp electrodes. Density (events/min)

	MD	Scalp	t	p
SO number	1408.5 \pm 126.2	1476.9 \pm 129.1	1.06	0.31
SO density	6.21 \pm 0.22	6.40 \pm 0.25	-0.5	0.61
SO-coupling [%]	30.45 \pm 3.02	29.51 \pm 3.15	1.37	0.35
spindle number	2127.8 \pm 238.2	1796.5 \pm 299.8	3.35	0.005
spindle density	9.81 \pm 0.98	6.14 \pm 1.04	2.43	0.024
spindle-coupling [%]	45.1 \pm 5.5	30.2 \pm 4.6	2.18	0.046
SO-spindle number	488 \pm 53.3	338.5 \pm 79.7	1.78	0.095

Minor points

6. Fig. 1: the description of the images in panel (a) is not clear. Are these real T1-weighted MRI images with superimposed red and blue areas, or else? Also, there is a grey shadow in panel (c) the meaning of which is not clear and not explained in the legend. Please, label appropriately the X axis of panel (b).

Response: We now add the following information to the legend of Figure 1:

-Localization of an exemplar patient's DBS leads³² is shown on a T1-weighted template MRI with superimposed thalamus atlas³³.

-The grey shadow indicates NREM sleep stages N2 and SWS. We added this information to the figure legend. We now appropriately labeled the X axis of panel (b).

Fig 1. | Thalamic electrode placement and sleep architecture. (a) Electrodes were implanted in the left and right ANT (blue) and MD (red). Localization of an exemplar patient's DBS leads [insert lead-dbs citation] is shown on a T1-weighted template MRI with superimposed thalamus atlas³³. (b) Example of NREM sleep segment (15 sec), comprising SOs and sleep spindles (top row: scalp recording; bottom row: ANT). (c) Hypnogram of a sample participant, showing time spent in different sleep stages across one recording night. The grey shading indicates NREM sleep stages N2 and SWS.

7. The results shown in Supplementary Figure 1 are very important: thus, they should be included in Figure 3.

Response: We included the results shown in Supplementary Figure 1 now both in the main results section as well as in Figure 3. We furthermore, added the preferred phase of ANT SOs for their coupling with MD SOs (Figure 3j).

The results section reads as follows (page 8, line 170):

...Finally, we directly assessed ANT – MD related SO interactions and tested whether ANT SOs would precede MD SOs (here occurrence probabilities were obtained for all possible ANT – MD combinations per participant, leading to 13 distributions). In line with the previous findings, results of this direct comparison revealed that ANT SOs preferentially emerge before MD SOs (positive cluster from -0.15 to 0 sec, $p = 0.015$; time of peak: -0.05 sec). Again, we obtained the phase of ANT SOs for all paired SO-events at the time of MD SO down state (± 750 ms). We found a significant nonuniform distribution across contacts (Rayleigh $z = 7.44$, $p = 0.0001$), with the phase of ANT SOs preceding their MD counterparts (mean coupling direction: $-138.34 \pm 11.08^\circ$; see Fig. 3j)...

Fig 3. | Neocortical SOs precede SOs in the MD. (a) Occurrence probabilities of MD SO down state peaks relative to neocortical SO down state peaks (time = 0, dashed line; bin size = 50 ms), illustrating that, on average, neocortical SOs lead MD SOs. The solid black line indicates significant differences, resulting from comparison with SO-free control events (positive cluster from -0.05 to 0.1 sec, $p = 0.021$; time of peak: 0.05 sec). (b) Phase of MD SOs at the time of neocortical SO downstate for paired SO-SO events, depicting that MD SO phases followed their neocortical counterparts (phase of neocortical SO down states corresponds to $\pm \pi$; mean coupling direction: $133.51 \pm 10.91^\circ$; Rayleigh test: $p < 0.0001$; $z = 7.94$). (c) Time–frequency representation of neocortical SOs (locked to neocortical SO down states), contrasted against event-free segments. The contour lines indicate clusters ($p < 0.05$, corrected for multiple comparisons across time and frequency). (d) Grand average EEG trace of neocortical SOs (mean \pm SEM, negative peak, time 0, $N = 1476.9 \pm 129.1$). (e) Time–frequency representation of all MD SOs (locked to MD SO down states), contrasted against event-free segments. The contour lines indicate clusters ($p < 0.05$, corrected for multiple comparisons across time and frequency). (f) Grand average iEEG trace of MD SOs (mean \pm SEM, negative peak, time 0, $N = 1408.5 \pm 126.2$). (g) Occurrence probabilities for the comparison of ANT-neocortical (blue) and MD-neocortical SO interactions (red), showing that ANT SOs preferentially emerge before MD SOs with regards to their neocortical counterparts (positive cluster from -0.25 to -0.05 sec, $p = 0.018$; corrected for multiple comparisons across time;). (h) The phase distribution for the coupling between thalamus and neocortex differed significantly when comparing ANT and MD related SO coupling (Watson-Williams test: $F = 12.7$, $p = 0.0015$). (i) Occurrence probabilities of ANT SO down state peaks relative to MD SO down state peaks, showing that in a direct comparison ANT SOs preferentially emerge before MD SOs (positive cluster from -0.15 to 0 sec, $p = 0.015$; time of peak: -0.05 sec). (j) Phase of ANT SOs at the time of MD SO downstate for paired SO-SO events, depicting that ANT SO phases preceded their MD counterparts (phase of MD SO down states corresponds to $\pm \pi$; mean coupling direction: $-138.34 \pm 11.08^\circ$; Rayleigh test: $p < 0.0001$; $z = 7.44$).

8. Pg 10 In 202: not clear what the meaning of “seamlessly” is in this sentence.

Response: We wanted to point out that oscillatory power associated with sleep spindles was not confined to the classical sleep spindle range (11-16 Hz), but instead was also visible in frequencies above 20 Hz. To prevent misunderstanding, we changed the respective wording (page 10, line 213).

...(~11-16 Hz; Fig. 4c), spindle – related power increases in case of ANT recordings were also evident in higher frequencies (> 20 Hz; Fig. 4e; $p < 0.05$, corrected for multiple comparisons across time and frequency)...

9. Pg 15, In 291: please abbreviate “phase-slope index” here.

Response: We now abbreviate the phase-slope-index (PSI) in line 291.

10. Pg 17 In 340: “significance” should be “significant”.

Response: We corrected the mistake.

Reviewer #3 (Remarks to the Author):

Comment 1: The finding that only SOs in ANT (but not in MD) lead neocortical SOs, whereas thalamic sleep spindles preceded their neocortical counterparts in both thalamic nuclei are interesting: Could these differences indicate a functional diversity not only of the thalamic nuclei but also of the sleep oscillations originating there?

Response: Yes, we fully agree with the reviewer. Our finding that only ANT SOs preceded neocortical SOs, while sleep spindles always led their neocortical counterparts irrespective of thalamic nucleus (i.e., ANT or MD) is in our view a clear sign for the functional diversity of thalamic nuclei and sleep oscillations originating there, pointing to a specific role of the ANT in the organization of SOs. To make this conclusion clearer we added a dedicated statement to the discussion (page 20, line 400):

...Our findings thus highlight both a specific role of the ANT for SO dynamics and, more generally, the functional diversity of specific thalamic nuclei in humans³⁴, also with regards to sleep oscillations originating there...

Comment 2: How can the finding that sleep spindles lock to the same very early phases of SOs in both thalamic nuclei and precede neocortical counterparts be reconciled with the differential orchestrations of SOs between ANT and neocortex vs. MD and neocortex? Are different anatomical loops involved? Fewer/more events coupled? Does this tell us something about the functional diversity of these sleep oscillations?

Response: This is an interesting point. Spindles in the ANT and the MD started at early phases of the SO (i.e., around the SO down-state; please see our response to Comment 3 by Reviewer #1, on how thalamic down-states are coupled to thalamic sleep spindles). However, as we point out in the discussion (see page 23, line 506), spindles in the MD seemed to start at even earlier phases with respect to MD SOs as compared to ANT spindles do with respect to ANT SOs (note that the direct comparison between MD and ANT did not reach significance ($p = 0.11$), which might be related to the modest sample-size; see Figure 7 h + i). Importantly, these observations fit very well to a systems perspective, where, in the cortex, sleep spindles should nest towards neocortical up-states to ensure that neocortical target areas are optimally tuned for synaptic plasticity and memory reprocessing^{35,36}. If ANT SOs and spindles are fundamental for the coordination of neocortical SO-spindle complexes, they would need to satisfy several conditions:

(1) ANT SOs and spindles need to precede their neocortical counterparts. Our data meets these predictions, with SOs in the ANT preceding neocortical SOs on average by 50 ms and sleep spindles preceding their neocortical counterparts on average by 100 ms.

(2) Given that ANT spindles tended to precede neocortical spindles to an even greater extent as it was the case for SOs (100 ms vs 50 ms), the observation that spindles in the ANT started at earlier SO phases (as compared to neocortex), is a necessary prerequisite to allow neocortical spindles to couple to neocortical SO up-states. Please see below (Comment 5) for a schematic overview of sleep oscillations related interactions between the ANT and the neocortex.

(3) The necessity for thalamic spindles to start at early SO phases seems even more important for MD related spindles. In the MD, a portion of neocortical SOs preceded MD SOs. In order for sleep spindles to arrive in the upstate of a neocortical SO, MD sleep spindles have to emerge relatively earlier with regards to the SO phase (and also in comparison to the ANT).

While we could not find a difference with regards to the SO-spindle coupling rate between the ANT and the MD ($t = 0.6057$, $p = 0.55$), these results further indicate the functional diversity of sleep related oscillations and the implicated thalamic nuclei.

We rephrased several parts of the discussion and hope that our reasoning becomes clearer now (please also see our related response to comment 5 and the newly added illustration below). The section reads as follow (page: 23, line: 494):

...Thus, ANT spindles started at significantly earlier ANT SO phases as compared to neocortical SO-spindle interactions. Notably, this divergent property of SO-spindle coordination is a necessary prerequisite to enable spindles (or rather their activity peaks) to arrive during the presence of SO up-states on neocortical sites. Specifically, while we show that both ANT SOs and spindles preceded their neocortical counterparts, the time lag between ANT and neocortical spindles exceeded the time lag between ANT and neocortical SOs (100 ms vs. 50 ms). Hence, to enable the prototypical SO-spindle modulation in neocortical circuits (i.e., nesting of spindles towards the SO upstate) under the assumption that both graph-elements in the neocortex are governed by the ANT, it is inevitable that spindles start at relatively earlier phases of the SO in the ANT as compared to spindles of neocortical SO-spindle complexes. Without such a shifted coordination on the thalamic level, cortical SO-spindle coupling would be systematically misaligned, with spindles regularly peaking after SO up-states.

We found a similar pattern of SO-spindle interactions within the MD, with spindles starting at early phases of the SO. Interestingly, even though not significant, spindles in the MD seem to start even earlier with respect to SOs as it is the case in the ANT. Again, in order to project spindles to the neocortex that nest in neocortical up states in a timely manner, a shifted coordination of SOs and spindles in the MD is necessary (i.e., spindles starting at early SO phases). This seems to be even more relevant in the case of the MD, where SOs were preceded (and potentially driven) by neocortical SOs, while spindles preceded neocortical spindles. Hence, for MD spindles to arrive at the neocortex during the presence of an SO-upstate, it is inevitable to start at an early MD SO-phase....

We added this information to the results section, page 17, line 356:

...There was no difference in the SO-spindle coupling rate between ANT and MD ($t = 0.6057$, $p = 0.55$)...

With respect to different anatomical loops being involved, not much is known in humans. Insights from animal models are available, but equivocal. For spindle generation, prominent models describe recurrent loops between thalamocortical neurons and the reticular nucleus (TRN). The projections between reticular nucleus and thalamic nuclei, however, differ with respect to the animal model used (see, e.g.,¹¹ for differences between felines and rodents). In the case of the ANT, Pare and colleagues⁹ even concluded that it is devoid of inputs from TRN (but see, e.g.,³⁷ for opposing evidence). Since we clearly identify sleep spindles in the ANT and also find coupling with neocortical spindles and thalamic as well as neocortical SOs, we assume that recurrent loops between ANT thalamocortical neurons and TRN exist and generate ANT spindles, similar to what has been shown in animal models^{7,8}. The finding of local spindles in ANT and MD (i.e., spindles that only occur in one but not the other thalamic nucleus) can be partly reconciled anatomically. On the one hand, TRN projections are organized topographically, with particular sectors of the TRN projecting to specific thalamic nuclei³⁷. On the other hand, it has been shown that TRN sectors projecting to the MD also project to ANT¹²⁻¹⁴. These overlapping TRN projections may be explained by the fact that MD and ANT thalamocortical neurons share the prefrontal cortex as common target.

We added a section to the discussion on page 22, line 467:

...With respect to the neural circuitry generating spindles in ANT and MD, one could assume recurrent loops between TRN and thalamocortical neurons, similar to what has been shown in animals^{7,8}. Extrapolating these findings from animals to humans needs to be done with caution, however, since result from animal models concerning TRN innervation of thalamic nuclei is not always univocal⁹. In general, it is assumed that TRN is organized in sectors that topographically project to thalamic nuclei¹⁰, which can explain why local spindles emerge in both MD and ANT. However, the mapping of TRN sections to thalamic nuclei is not one-to-one¹¹: for example, some TRN sectors projecting to the MD also project to ANT¹²⁻¹⁴. Such

overlapping projections seem plausible when considering that MD and ANT thalamocortical neurons share common targets, for example in the prefrontal cortex...

Comment 3: ll.252ff The logic of this statement and how it follows from the results reported so far was unclear to me. Do the authors mean that SOs leading in thalamus ANT/neocortex and SOs leading in neocortex for MD/neocortex indicates that both the neocortex and the thalamus can coordinate/initiate sleep oscillations? Could they please elaborate/clarify.

Response: It appears as if ll.252ff might not refer to the issue pointed out by the reviewer. Nevertheless, we hope to answer the (general) question as good as possible. On the basis of our data, we cannot exclude that SOs are likewise initiated in the cortex. Our result that only ~30 % of neocortical SOs were coupled to ANT SOs (and vice versa; see Supplementary Table 3) might speak for such an additional neocortical coordination. However, the novel finding in our work is that a good portion of neocortical SOs was putatively coordinated by ANT SOs, which goes against the traditional view of a cortical-only generation of SOs. That said, a pressing question for future research will be to delineate whether ANT coordinated SOs exhibit special features in the context of memory reprocessing during sleep. This might be conceivable due to their precise coupling to thalamic spindles, given that the precision of the coupling between SOs and spindles has been shown to be instrumental for memory consolidation to unfold (e.g., ^{36,38}). It could also be that spindles from ANT coordinated SO-spindle complexes are more efficient in grouping hippocampal ripples, due to the direct innervation of the hippocampal formation by the ANT ³⁹, which would indicate that the ANT facilitates the full hierarchy of sleep oscillations (SO-spindle-ripple complexes) that is thought to mediate the memory function of sleep ⁴⁰. In this regard, our work represents a starting point for elucidating the distinct role of the ANT in coordinating sleep oscillations and memory consolidation.

We added a section to the discussion on page 21, line 438, to clarify this:

...We here show ANT SOs leading neocortical SOs on the basis of approximately 30% of neocortical SOs that were coupled to ANT SOs. Neocortical SOs that were not coupled to the ANT may as well have been initiated in the neocortex, in line with traditional accounts of the origin of SOs^{20,41}. Future research needs to investigate whether ANT coordinated SOs are functionally distinct from neocortically coordinated SOs, for example with respect to memory reprocessing during sleep. The precise coupling between SOs and thalamic spindles might be of relevance here, given that the precision of the coupling between SOs and spindles has been shown to be instrumental for memory consolidation to unfold (e.g., ^{36,38}). Due to its prominent location within the limbic circuit and direct projections to the hippocampal formation³⁹, the ANT could also be important for integrating hippocampal ripples into SO-spindle complexes, facilitating the full triple-coupling of sleep oscillations that is assumed to mediate the memory functions of sleep⁴⁰. Hence, our work represents a starting point for elucidating the distinct role of the ANT in coordinating sleep oscillations and memory consolidation...

Comment 4: ll. 381ff The data reported here show that both ANT SOs and spindles lead those in the neocortex. SO-spindle interactions are deemed central to offline memory processing and plasticity during sleep. The ANT has extensive connections to neocortical areas which have previously been perceived to a) be the origin of neocortical SOs and b) are critically involved in memory functions. The data reported in this paper suggest that the SOs detected in central mnemonic processing hubs of the neocortex may instead originate in the thalamus which may have strong implications for its role in coordinating memory reprocessing and storage in the human brain. This is a central point of discussion and could be phrased even more strongly, in my opinion. The results clearly encourage further investigation of a critical role of the thalamus, and ANT in particular, in memory functions.

Response: We completely agree with the reviewer that our findings might have strong implications for the role of the thalamus (and specifically the ANT) in coordinating memory consolidation during sleep. We also tried to make this point clear in the discussion. However, given that the current manuscript

lacks behavioral data, which would be necessary to draw strong conclusions concerning the role of the ANT in governing memory reprocessing during sleep, we wanted to be cautious with strong statements in this direction. That said, we added multiple sentences to the discussion, specifically referring to the potential role of the ANT in memory consolidation.

Line 523: Taken together, our findings provide first evidence for the ANT as a major hub for coordinating the cardinal NREM-sleep related oscillations. In the neocortex, both spindles and SOs were led by ANT-related activity, while the interplay of local ANT SO-spindles was tuned in a way that allows for neocortical spindles to nest within SO up states. Importantly, this exact coordination is thought to be instrumental for the consolidation of memories⁴². Hence, the ANT might represent a key-player in governing memory reprocessing during human NREM sleep.

Line 541: Our data critically extend these theoretical considerations, by spotlighting the human thalamus, and in specific the ANT, as a putative active agent in interfacing sleep-related oscillations between brain regions and thereby, potentially facilitating memory consolidation.

Line 544: While our current data remain agnostic with regards to hippocampal activity, it has been shown in a single patient study, that hippocampal spindles are aligned with SOs in the ANT⁴³. Hence, ANT related oscillations might not only migrate to the neocortex but likewise to the hippocampus, where spindles are known to govern ripples and associated memory reactivation⁴⁴. Taken together, ANT related activity might critically contribute to the triple coupling of sleep oscillations and thus, the memory function of sleep. Future work will need to capture the association of ANT-neocortical interactions with the behavioral expressions of memory consolidation, as well as continue characterizing the functionally diverse human thalamus. Future work relating the association of ANT-neocortical interactions with the behavioral expressions of memory consolidation will elucidate the memory function of the ANT and further characterize the functionally diverse human thalamus.

Please also see our response to the comment 3, where we now include speculations about the functional role of ANT coordinated SOs.

Comment 5: ll. 410 and following paragraph: The reasoning why timing of sleep spindles in the SO downstate in thalamus vs. down-to-up-state/up-state in neocortex indicates a concomitant grouping of SOs and spindles by the ANT was not immediately accessible to me (see also l. 421, ll.425ff). Why is timing not conserved? Why is timing different in ANT-neo vs. MD-neo? How can this be reconciled? What implications may this have for functional divergence of MD and ANT in the coordination of sleep oscillations?

Response: We thank the reviewer for the opportunity to clarify.

The precise coupling of SOs and spindles on a neocortical level, i.e., sleep spindles peaking during SO up-states, has been shown to be an important pre-requisite for facilitating synaptic plasticity and by that allowing long-lasting memory representations to be established in cortical circuits^{36,45}. To reconcile the timing of this coupling in the neocortex with our observed timing differences of spindles and SOs in the ANT versus the cortex, the coupling of spindles and SOs in the ANT need to divert from the timing of SO-spindle coupling in the neocortex.

In our data, we observed SOs in the ANT preceding neocortical SOs on average by 50 ms and sleep spindles preceding their neocortical counterparts on average by 100 ms. Therefore, in order to allow for the above described precise coupling of SOs and spindles in the cortex, spindles in the ANT need to start at earlier SO phases (as compared to neocortex). Without such a shifted coordination on the thalamic level, neocortical SO-spindle coupling would be systematically misaligned, with spindles regularly peaking after SO up-states. In the MD, sleep spindles have to emerge even earlier with respect to the SO phase to reach the neocortex during the presence of SO related up-states, as MD SOs were preceded by cortical SOs, while MD spindles likewise preceded neocortical spindles.

We hope that including a schematic figure on the timing and coupling of ANT versus neocortical oscillations (Fig. 8; see below) will improve the understanding our rationale.

As outlined in the response to comment 2, we re-phrased the section related to SO-spindle coupling in the discussion, in the hope that our reasoning becomes more comprehensible.

The section reads as follow (page: 23, line: 494):

...Thus, ANT spindles started at significantly earlier ANT SO phases as compared to neocortical SO-spindle interactions. Notably, this divergent property of SO-spindle coordination is a necessary pre-requisite to enable spindles (or rather their activity peaks) to arrive during the presence of SO up-states on neocortical sites. Specifically, while we show that both ANT SOs and spindles preceded their neocortical counterparts, the time lag between ANT and neocortical spindles exceeded the time lag between ANT and neocortical SOs (100 ms vs. 50 ms). Hence, to enable the prototypical SO-spindle modulation in neocortical circuits (i.e., nesting of spindles towards the SO upstate) under the assumption that both graph-elements in the neocortex are governed by the ANT, it is inevitable that spindles start at relatively earlier phases of the SO in the ANT as compared to spindles of neocortical SO-spindle complexes. Without such a shifted coordination on the thalamic level, cortical SO-spindle coupling would be systematically misaligned, with spindles regularly peaking after SO up-states.

We found a similar pattern of SO-spindle interactions within the MD, with spindles starting at early phases of the SO. Interestingly, even though not significant, spindles in the MD seem to start even earlier with respect to SOs as it is the case in the ANT. Again, in order to project spindles to the neocortex that nest in neocortical up states in a timely manner, a shifted coordination of SOs and spindles in the MD is necessary (i.e., spindles starting at early SO phases). This seems to be even more relevant in the case of the MD, where SOs were preceded (and potentially driven) by neocortical SOs, while spindles preceded neocortical spindles. Hence, for MD spindles to arrive at the neocortex during the presence of an SO-upstate, it is inevitable to start at an early MD SO-phase....

Could the authors imagine adding a figure that illustrates the interactions they propose to underlie these differential oscillatory dynamics, and how coordination by ANT follows from the observed differences in timing of oscillatory interactions? If these is beyond scope, could the authors add additional information on why the observed timing of oscillations is a necessary pre-requisite to allow for concomitant coordination of neocortical SOs and sleep spindles by ANT?

Response: We tried to make this point clearer in text (please see responses above). In addition, as suggested by the reviewer, we made an additional figure illustrating the temporal relationship of thalamo-cortical oscillations (Figure 8 in the manuscript).

We refer to the new Figure in the discussion (page 23 line 506):

...Without such a shifted coordination on the thalamic level, cortical SO-spindle coupling would be systematically misaligned, with spindles regularly peaking after SO up-states (see Fig. 8 for a schematic illustration)...

Fig. 8 | Schematic overview of sleep oscillations related interactions between the ANT and the neocortex: (a) ANT SOs (bottom, blue) that are coupled to neocortical SOs, preferentially peak on average ~50 ms before neocortical SOs (top, grey). (b) ANT spindles (bottom, blue) that are coupled to neocortical spindles, peak on average 100 ms before neocortical spindles (top, grey). (c) To enable the prototypical SO-spindle modulation in neocortical circuits (i.e., nesting of spindles towards the SO upstate; top in grey), ANT spindles start at significantly earlier SO phases as compared to neocortical SO-spindle interactions (i.e., spindles in the ANT start closer to the ANT SO down-state, while spindles in the neocortex start at the transition from down- to up-state). The insets illustrate the difference in the phase locking of spindle start to their respective SO (the phase of SO down states corresponds to $\pm\pi$).

The following parts would, in my view, benefit from further clarification:

- II. 414: “Hence, the characteristics of SO-spindle interactions in the ANT might therefore highlight how the thalamus concomitantly groups SOs and spindles in the neocortex, leading to the well-known oscillatory nesting.”

Response: page 22, line 487:

...Hence, the characteristics of SO-spindle interactions in the ANT might highlight how the thalamus **simultaneously** groups SOs and spindles in the neocortex, leading to the well-known oscillatory nesting (**i.e., spindles nesting towards SO up-states**)...

- II. 421ff “Notably, this divergent property of SO-spindle coordination is a necessary pre-requisite to allow for a concomitant coordination of neocortical SOs and spindles by the ANT. Specifically, while we show that both ANT SOs and spindles preceded their neocortical counterparts, the time lag between ANT and neocortical spindles exceeded the time lag between ANT and neocortical SOs. Hence, to enable the prototypical SO-spindle modulation in neocortical circuits (i.e., nesting of spindles towards the SO upstate) under the assumption that both graph-elements in the neocortex are governed by the ANT, it is inevitable that spindles start at relatively earlier phases of the SO in the ANT as compared to spindles of neocortical SO-spindle complexes.”

Response: We rephrased the section (page 23, line 494):

...Thus, ANT spindles started at significantly earlier ANT SO phases as compared to neocortical SO-spindle interactions. Notably, this divergent property of SO-spindle coordination in the ANT is a consequence of the delays between ANT and cortical sleep oscillations we found. It allows **spindles (or, rather, their activity peaks), putatively generated in the ANT, to arrive during SO up-states on neocortical sites, generating the prototypical SO-spindle coupling found in scalp EEG**. Specifically, while we show that both ANT SOs and spindles preceded their neocortical counterparts, the time lag between ANT and neocortical spindles exceeded the time lag between ANT and neocortical SOs (100 ms vs. 50 ms). Hence, to enable the prototypical SO-spindle modulation in neocortical circuits (i.e., nesting of spindles towards the SO upstate) under the assumption that both graph-elements in the neocortex are governed by the ANT, it

is inevitable that spindles start at relatively earlier phases of the SO in the ANT as compared to spindles of neocortical SO-spindle complexes. Without such a shifted coordination on the thalamic level, cortical SO-spindle coupling would be systematically misaligned, with spindles regularly peaking after SO up-states (see Fig. 8 for a schematic illustration)...

- II 433ff “Again, in order to project spindles to the neocortex that nest in neocortical up states in a timely manner, a shifted coordination of SOs and spindles in the MD is necessary.”

Response: page 23, line 511:

...Again, in order to project spindles to the neocortex that nest in neocortical up states in a timely manner, a shifted coordination of SOs and spindles in the MD is necessary (i.e., spindles starting at early SO phases). This seems to be even more relevant in the case of the MD, where SOs were preceded (and potentially driven) by neocortical SOs, while spindles preceded neocortical spindles. Hence, for MD spindles to arrive at the neocortex during the presence of an SO-upstate, it is inevitable to start at an early MD SO-phase...

- II. 438ff “while the interplay of local ANT SO-spindles was tuned in a way that allows for neocortical spindles to nest within SO up states. Importantly, this exact coordination is thought to be instrumental for the consolidation of memories.”

Response: page 23, line 524:

... In the neocortex, both spindles and SOs were led by ANT-related activity, while the interplay of local ANT SO-spindles was tuned in a way that allows for neocortical spindles to nest within SO up states (ANT spindles starting at early ANT SO phases, see Fig. 8)...

Comment 6: II. 414 Stylistic comment: “Hence..., ...therefore...” one of the adverbs could be removed.

Response: We removed the word therefore.

Comment 7: Oscillation detection: Does sleep spindle detection work equally well in scalp and thalamic recordings? Have the same thresholds been applied/were approx. the same numbers of events detected? Are slopes and amplitudes comparable? Could ease of detection systematically influence the estimated timing of spindle and SO phase?

Response: We want to thank the reviewer for this thoughtful question. The same procedures for detecting spindles in both scalp and thalamic recordings have been applied following a common rationale (please see *spindle detection* in the Methods section, line 659, page 28):

The root mean square (RMS) signal was calculated using a moving average of 200 ms, and spindles that exceeded the mean amplitude of all detected events by 1.5 SD entered the analysis. Whenever the signal exceeded this threshold for more than 0.5 sec but less than 3 sec (duration criteria), a spindle event was detected.

As shown in Supplementary Tables 3 and 4 overall comparable numbers of sleep spindles were detectable in scalp (~1900) and thalamic recordings (ANT: 2203; MD: 2127). That said, the number of detected spindles differed significantly between thalamic and scalp EEG recordings (ANT: $p = 0.008$; MD: $p = 0.005$), with a higher number of spindles being detectable in thalamic recordings. This difference probably has multiple reasons, among them a poorer signal-to-noise ratio in scalp EEG as compared to intracranial recordings and potentially less mixing of neural sources in intracranial recordings (with bipolar reference) than scalp EEG. Additionally, spindles are generated in a thalamus-

TRN circuit [cite e.g., Fernandez & Lüthi] with complex projection properties [Gonzalo-Ruiz, A. & Lieberman], resulting in local and non-local thalamic spindles. Due to similar cortical projection sites (e.g., in the prefrontal cortex), a direct comparison of spindle numbers recorded in the thalamus and via scalp EEG is limited in its informative value.

Following the reviewer’s suggestion, we further set out to test whether there would be any differences in terms of amplitudes (spindles and SOs) and slopes (SOs) between scalp and thalamic-detected events (in Supplementary Tables 4 and 5 we already report that SO number did not differ between recording sites).

As shown in the tables below, spindle amplitude did not differ between events detected on the scalp and thalamic level (both, ANT and MD). Likewise, neither SO-amplitudes nor SO-slopes (i.e., down-to-upstate transition in $\mu\text{V}/\text{second}$) differed between recording sites. That said, it has to be noticed that comparing amplitudes between recording sites is due to differing re-referencing strategies difficult (i.e., common average in case of scalp EEG and bipolar referencing in case of thalamic contacts). Hence, we would refrain from including amplitude related metrics to the Supplementary information. We added the SO slope related information to Supplementary table 5 (also see below).

Oscillations ANT: Data are means \pm s.e.m. Spindle amplitudes, SO amplitudes and SO slopes (down-to-up transition) of paired events in NREM sleep at ANT contacts and scalp electrodes.

	ANT	Scalp	t	P
Spindle amplitude [μV]	8.06 ± 0.71	7.36 ± 0.28	0.78	0.44
SO amplitude [μV]	38.46 ± 4.78	32.18 ± 3.76	0.94	0.35
SO slope [$\mu\text{V}/\text{s}$]	75.99 ± 9.61	67.55 ± 6.63	0.64	0.52

Oscillations MD: Data are means \pm s.e.m. Spindle amplitudes, SO amplitudes and SO slopes (down-to-up transition) of paired events in NREM sleep at ANT contacts and scalp electrodes.

	MD	Scalp	t	P
Spindle amplitude [μV]	8.62 ± 0.58	7.39 ± 0.81	1.24	0.22
SO amplitude [μV]	37.57 ± 3.98	36.80 ± 3.25	0.11	0.91
SO slope [$\mu\text{V}/\text{s}$]	74.82 ± 7.58	80.69 ± 12.93	-0.42	0.67

Supplementary Table 5 | SO features: Data are means \pm s.e.m. Duration and relative occurrence during N2 and N3 sleep for scalp (ANT and MD analyses related), ANT and MD derived SOs.

SO	Duration [sec]	Slope [$\mu\text{V}/\text{sec}$]	Rel. N2 [%]	Rel. N3 [%]
scalp ANT	1.31 ± 0.01	67.55 ± 6.63	46.56 ± 3.79	53.43 ± 3.79
scalp MD	1.35 ± 0.02	80.69 ± 12.93	49.21 ± 4.38	50.78 ± 4.38
ANT	1.30 ± 0.01	75.99 ± 9.61	46.05 ± 3.87	53.94 ± 3.87
MD	1.30 ± 0.02	74.82 ± 7.58	47.70 ± 4.89	52.29 ± 4.89

Comment 8: Is the finding that thalamic spindles (in both ANT and MD) seamlessly transition from the “traditional” frequency range to higher frequencies an issue of power/filtering of spindle oscillations by the skull or a result from different network properties/functional diversity of brain regions?

Response: This is an interesting question and at this point we can only speculate about potential answers. It is true that intracranial recordings exhibit a higher signal to noise ratio as compared to scalp

EEG recordings (with an SNR of up to 100 times higher⁴⁶). In addition, as mentioned by the reviewer the properties of the skull dampen higher frequencies in scalp EEG recordings⁴⁷.

To test whether the higher frequencies observed in intracranial (thalamic) spindles are an issue of power/filtering by the skull, we detected sleep spindles in frontal intracranial contacts in three pre-surgical epilepsy patients (frontal iEEG contacts, n = 13; for details please see the response to comment 1 from reviewer 1).

The Figure below shows in (a) the time frequency representation of sleep spindles in the ANT; and in (b) the time frequency representation of sleep spindles as identified in frontal intracranial contacts. Spindle – related power increases were also evident in in higher frequencies of the frontal iEEG recordings. Hence, we conclude that these effects are rather a consequence of the recording technique (i.e., higher SNR for iEEG recordings, and no dampening of higher frequencies) and do not speak for distinct network properties/functional diversity of brain regions.

References:

1. Lachaux, J. P., Rudrauf, D. & Kahane, P. Intracranial EEG and human brain mapping. in *Journal of Physiology Paris* (2003). doi:10.1016/j.jphysparis.2004.01.018
2. Marmor, O. et al. Local vs. Volume conductance activity of field potentials in the human subthalamic nucleus. *J. Neurophysiol.* (2017). doi:10.1152/jn.00756.2016
3. Wennberg, R. & Lozano, A. M. Restating the importance of bipolar recording in subcortical nuclei [1]. *Clinical Neurophysiology* (2006). doi:10.1016/j.clinph.2005.09.020
4. Wennberg, R. A. & Lozano, A. M. Intracranial volume conduction of cortical spikes and sleep potentials recorded with deep brain stimulating electrodes. *Clin. Neurophysiol.* **114**, 1403–1418 (2003).
5. McCormick, D. A. & Bal, T. Sleep and arousal: Thalamocortical mechanisms. *Annual Review of Neuroscience* (1997). doi:10.1146/annurev.neuro.20.1.185
6. Bastuji, H., Lamouroux, P., Villalba, M., Magnin, M. & Garcia-Larrea, L. Local sleep spindles in the human thalamus. *J. Physiol.* **598**, 2109–2124 (2020).
7. Steriade, M. Grouping of brain rhythms in corticothalamic systems. *Neuroscience* **137**, 1087–1106 (2006).
8. Fernandez, L. M. J. & Lüthi, A. Sleep Spindles: Mechanisms and Functions. *Physiological reviews* **100**, 805–868 (2020).
9. Pare, D., Steriade, M., Deschenes, M. & Oakson, G. Physiological characteristics of anterior thalamic nuclei, a group devoid of inputs from reticular thalamic nucleus. <https://doi.org/10.1152/jn.1987.57.6.1669> **57**, 1669–1685 (1987).
10. Gonzalo-Ruiz, A. & Lieberman, A. R. Topographic organization of projections from the thalamic reticular nucleus to the anterior thalamic nuclei in the rat. *Brain Res. Bull.* **37**, 17–35 (1995).
11. Guillery, R. W., Feig, S. L. & Lozsádi, D. A. Paying attention to the thalamic reticular nucleus.

- Trends Neurosci.* (1998). doi:10.1016/S0166-2236(97)01157-0
12. Kultas-Ilinsky, K., Yi, H. & Ilinsky, I. A. Nucleus reticularis thalami input to the anterior thalamic nuclei in the monkey: a light and electron microscopic study. *Neurosci. Lett.* (1995). doi:10.1016/0304-3940(95)11273-Y
 13. Tai, Y., Yi, H., Ilinsky, I. A. & Kultas-Ilinsky, K. Nucleus reticularis thalami connections with the mediodorsal thalamic nucleus: A light and electron microscopic study in the monkey. *Brain Res. Bull.* (1995). doi:10.1016/0361-9230(95)02018-M
 14. Ilinsky, I. A., Ambardekar, A. V. & Kultas-Ilinsky, K. Organization of projections from the anterior pole of the nucleus reticularis thalami (NRT) to subdivisions of the motor thalamus: Light and electron microscopic studies in the rhesus monkey. *J. Comp. Neurol.* (1999). doi:10.1002/(SICI)1096-9861(19990705)409:3<369::AID-CNE3>3.0.CO;2-H
 15. Steriade, M., McCormick, D. A. & Sejnowski, T. J. Thalamocortical oscillations in the sleeping and aroused brain. *Science* **262**, 679–685 (1993).
 16. Mak-McCully, R. A. *et al.* Coordination of cortical and thalamic activity during non-REM sleep in humans. *Nat. Commun.* **8**, 15499 (2017).
 17. Helfrich, R. F. *et al.* Bidirectional prefrontal-hippocampal dynamics organize information transfer during sleep in humans. *Nat. Commun.* (2019). doi:10.1038/s41467-019-11444-x
 18. Mölle, M., Bergmann, T. O., Marshall, L. & Born, J. Fast and slow spindles during the sleep slow oscillation: disparate coalescence and engagement in memory processing. *Sleep* **34**, 1411–21 (2011).
 19. Muehlroth, B. E. *et al.* Precise Slow Oscillation–Spindle Coupling Promotes Memory Consolidation in Younger and Older Adults. *Sci. Rep.* **9**, 1–15 (2019).
 20. Timofeev, I., Grenier, F., Bazhenov, M., Sejnowski, T. J. & Steriade, M. Origin of slow cortical oscillations in deafferented cortical slabs. *Cereb. Cortex* (2000). doi:10.1093/cercor/10.12.1185
 21. Murphy, M. *et al.* Source modeling sleep slow waves. *Proc. Natl. Acad. Sci. U. S. A.* **106**, 1608–13 (2009).
 22. Shibata, H. & Kato, A. Topographic relationship between anteromedial thalamic nucleus neurons and their cortical terminal fields in the rat. *Neurosci. Res.* (1993). doi:10.1016/0168-0102(93)90030-T
 23. Van Groen, T., Kadish, I. & Wyss, J. M. Efferent connections of the anteromedial nucleus of the thalamus of the rat. *Brain Research Reviews* (1999). doi:10.1016/S0165-0173(99)00006-5
 24. Child, N. D. & Benarroch, E. E. Anterior nucleus of the thalamus: functional organization and clinical implications. *Neurology* **81**, 1869–1876 (2013).
 25. Jankowski, M. M. *et al.* The anterior thalamus provides a subcortical circuit supporting memory and spatial navigation. *Front. Syst. Neurosci.* **7**, (2013).
 26. Liguori, C., Toledo, M. & Kothare, S. Effects of anti-seizure medications on sleep architecture and daytime sleepiness in patients with epilepsy: A literature review. *Sleep Medicine Reviews* (2021). doi:10.1016/j.smr.2021.101559
 27. Frauscher, B. *et al.* Facilitation of epileptic activity during sleep is mediated by high amplitude slow waves. *Brain* (2015). doi:10.1093/brain/awv073
 28. Frauscher, B., von Ellenrieder, N., Dubeau, F. & Gotman, J. Scalp spindles are associated with widespread intracranial activity with unexpectedly low synchrony. *Neuroimage* (2015). doi:10.1016/j.neuroimage.2014.10.048
 29. Steriade, M. Sleep, epilepsy and thalamic reticular inhibitory neurons. *Trends in Neurosciences* (2005). doi:10.1016/j.tins.2005.03.007
 30. Simon, K. C. *et al.* Slow oscillation density and amplitude decrease across development in pediatric Duchenne and Becker muscular dystrophy. *Sleep* (2021). doi:10.1093/sleep/zsaa240
 31. Gais, S., Mölle, M., Helms, K. & Born, J. Learning-dependent increases in sleep spindle density. *J. Neurosci.* **22**, 6830–4 (2002).
 32. Horn, A. & Kühn, A. A. Lead-DBS: a toolbox for deep brain stimulation electrode localizations and visualizations. *Neuroimage* **107**, 127–135 (2015).

33. Ewert, S. *et al.* Toward defining deep brain stimulation targets in MNI space: A subcortical atlas based on multimodal MRI, histology and structural connectivity. *Neuroimage* **170**, 271–282 (2018).
34. Halassa, M. M. & Sherman, S. M. Thalamocortical Circuit Motifs: A General Framework. *Neuron* **103**, 762–770 (2019).
35. Seibt, J. *et al.* Cortical dendritic activity correlates with spindle-rich oscillations during sleep in rodents. *Nat. Commun.* **8**, (2017).
36. Helfrich, R. F., Mander, B. A., Jagust, W. J., Knight, R. T. & Walker, M. P. Old Brains Come Uncoupled in Sleep: Slow Wave-Spindle Synchrony, Brain Atrophy, and Forgetting. *Neuron* **97**, 221-230.e4 (2018).
37. Gonzalo-Ruiz, A. & Lieberman, A. R. Topographic organization of projections from the thalamic reticular nucleus to the anterior thalamic nuclei in the rat. *Brain Res. Bull.* (1995). doi:10.1016/0361-9230(94)00252-5
38. Schreiner, T., Petzka, M., Staudigl, T. & Staresina, B. P. Endogenous memory reactivation during sleep in humans is clocked by slow oscillation-spindle complexes. *Nat. Commun.* (2021). doi:10.1038/s41467-021-23520-2
39. Aggleton, J. P. & O'Mara, S. M. The anterior thalamic nuclei: core components of a tripartite episodic memory system. *Nat. Rev. Neurosci.* (2022). doi:10.1038/S41583-022-00591-8
40. Girardeau, G. & Lopes-dos-Santos, V. Brain neural patterns and the memory function of sleep. *Science (80-.).* **374**, 560–564 (2021).
41. Steriade, M., McCormick, D. A. & Sejnowski, T. J. Thalamocortical oscillations in the sleeping and aroused brain. *Science (80-.).* (1993). doi:10.1126/science.8235588
42. Klinzing, J. G., Niethard, N. & Born, J. Mechanisms of systems memory consolidation during sleep. *Nature Neuroscience* (2019). doi:10.1038/s41593-019-0467-3
43. Sarasso, S. *et al.* Hippocampal sleep spindles preceding neocortical sleep onset in humans. *Neuroimage* **86**, 425–432 (2014).
44. Kim, J., Gulati, T. & Ganguly, K. Competing Roles of Slow Oscillations and Delta Waves in Memory Consolidation versus Forgetting. *Cell* **179**, 514-526.e13 (2019).
45. Niethard, N., Ngo, H. V. V., Ehrlich, I. & Born, J. Cortical circuit activity underlying sleep slow oscillations and spindles. *Proc. Natl. Acad. Sci. U. S. A.* (2018). doi:10.1073/pnas.1805517115
46. Parvizi, J. & Kastner, S. Promises and limitations of human intracranial electroencephalography. *Nat. Neurosci.* **21**, 474–483 (2018).
47. Davidson, R. J., Jackson, D. C. & Larson, C. L. Human electroencephalography. in *Handbook of psychophysiology, 2nd ed.* 27–52 (Cambridge University Press, 2000).

REVIEWERS' COMMENTS

Reviewer #1 (Remarks to the Author):

The authors have addressed my comments in an overall quite satisfactory manner. In my view, the manuscript is acceptable after two minor revision are added, regarding my original comment #1 and #3.

Comment #1: Here, in order to examine whether the skull acting as a low pass filter, may have delayed the picked-up signal at the scalp level relative to thalamic recordings, the authors provide an additional data-set in 3 patients where the scalp EEG (from FP1) was recorded in close proximity to the intracranial recordings from frontal cortex. They report that the phase of intracranial SOs is “almost identical to the phase of scalp detected events”. Although these are very convincing data, I think they do not entirely exclude the possibility that the result of ANT SOs leading (by 50 ms) surface EEG SOs is biased by a technical artifact. First, the authors do not test significance of the difference in phase angles for EEG and intracranial SOs and if they did, this would be null hypothesis testing. And second, the lead of ANT SOs is small (50 ms) which corresponds to the bin size/resolution in the analyses; visual inspection of the Suppl. Fig 3 panel a indeed shows a small left-side shift of the distribution of intracranial SOs. Given these uncertainties, the authors should acknowledge in the Discussion section of the main text that due to the general restrictions of intracranial recordings in humans, it cannot be entirely excluded that the lead in ANT SOs with respect to surface EEG SOs is confounded through technical differences in recordings (intracranial vs scalp).

Comment #3. As far as I understand, the SO-spindle coupling as illustrated in panel a of the Figure shown in the rebuttal letter, refers to the spindle onsets only (whereas for the Mak-McCully data distributions of spindle onsets and offsets are shown. The authors should clearly indicate (in the legends to their respective figures) that the temporal distributions refer to spindle onsets (rather than to spindle maxima). Alternatively, the authors could present both distributions of spindle onsets and offsets, like in Mak-McCully et al.. (Please, change in Suppl. Fig. 3 b “ANT SOs” to, e.g., “intracortical SOs“.)

Reviewer #2 (Remarks to the Author):

The authors have satisfactorily addressed all my comments.

I have no further issues with this manuscript.

Reviewer #3 (Remarks to the Author):

I would like to thank the authors for their considerate and careful responses to my comments. All questions and issues I have raised have been clarified in the manuscript and I have no further remarks. Congratulations on this impressive work!

Comment #1: Here, in order to examine whether the skull acting as a low pass filter, may have delayed the picked-up signal at the scalp level relative to thalamic recordings, the authors provide an additional data-set in 3 patients where the scalp EEG (from FP1) was recorded in close proximity to the intracranial recordings from frontal cortex. They report that the phase of intracranial SOs is "almost identical to the phase of scalp detected events". Although these are very convincing data, I think they do not entirely exclude the possibility that the result of ANT SOs leading (by 50 ms) surface EEG SOs is biased by a technical artifact. First, the authors do not test significance of the difference in phase angles for EEG and intracranial SOs and if they did, this would be null hypothesis testing. And second, the lead of ANT SOs is small (50 ms) which corresponds to the bin size/resolution in the analyses; visual inspection of the Suppl. Fig 3 panel a indeed shows a small left-side shift of the distribution of intracranial SOs. Given these uncertainties, the authors should acknowledge in the Discussion section of the main text that due to the general restrictions of intracranial recordings in humans, it cannot be entirely excluded that the lead in ANT SOs with respect to surface EEG SOs is confounded through technical differences in recordings (intracranial vs scalp).

Response: We thank the reviewer for this thoughtful comment. We added an additional sentence to the discussion, acknowledging this potential limitation.

Page 13, line 365:

...To mitigate these concerns, we show, in an independent data set, that intracranially recorded SOs in the frontal lobe exhibit no temporal delay with regards to SOs simultaneously recorded by frontal scalp EEG (see Supplementary Fig. 3). However, we acknowledge that a bias in the relative timing between ANT and scalp SOs cannot be completely excluded due to technical differences in recording techniques (intracranial vs. scalp EEG)....

Comment #3. As far as I understand, the SO-spindle coupling as illustrated in panel a of the Figure shown in the rebuttal letter, refers to the spindle onsets only (whereas for the Mak-McCully data distributions of spindle onsets and offsets are shown. The authors should clearly indicate (in the legends to their respective figures) that the temporal distributions refer to spindle onsets (rather than to spindle maxima). Alternatively, the authors could present both distributions of spindle onsets and offsets, like in Mak-McCully et al.. (Please, change in Suppl. Fig. 3 b "ANT SOs" to, e.g., "intracortical SOs")

Response: The information that the temporal distributions refer to spindle onsets (rather than to spindle maxima) in case of SO-spindle events is already available in the legends of Figures 6 & 7 and similarly explicated in the results and methods sections. We highlight the corresponding sections of the figure legends and results on the following pages and hope that this information is sufficient.

We corrected the mistake in the legend of Supplementary Figure 3b.

Figure legends:

Figure 6. ANT spindles lock to early SO phases. (a) Occurrence probabilities of **neocortical spindle onsets** with respect to neocortical SO down states (bin size = 50 ms), illustrating that **spindles started at the SO down-to-up transition** (0.05 to 0.15 sec; dependent-samples t-test, two-sided, $p < 0.0001$; cluster corrected across time; peak: 0.1 sec). (b) **ANT spindles emerged around ANT SO down states** (-0.15 to 0.25 sec; dependent-samples t-test, two-sided, $p = 0.0012$; cluster corrected across time; peak: 0 sec). (c) Time–frequency representation (TFR) of neocortical SO-spindle events ($N = 304.1 \pm 69.8$; locked to neocortical SO down states), contrasted against event-free segments. The contour lines indicate clusters (dependent-samples t-test, two-sided, $p < 0.05$, cluster corrected across time and frequency). (d) TFR of all ANT SO-spindle events ($N = 431 \pm 52.1$, locked to ANT SO down states), contrasted against event-free segments. The contour lines indicate clusters (dependent-samples t-test, two-sided, $p < 0.05$, cluster corrected across time and frequency). (e) Comparing the occurrence probabilities of ANT (blue) and neocortical SO-spindle complexes (gray) yielded that ANT spindles locked significantly earlier to SOs as compared to neocortical spindles (-0.3 to -0.1, dependent-samples t-test, two-sided, $p = 0.0039$; cluster corrected across time). (f) Phases of the SO-spindle modulation derived from neocortical (gray) and ANT (blue) events. **Neocortical spindle onsets emerged** at the neocortical SO down-to-up transition (neocortical SO down state phase corresponds to $\pm \pi$; mean coupling direction: $-137.14 \pm 7.58^\circ$; Rayleigh test, one-sided: $z = 6.91$, $p < 0.0001$), **while ANT spindles started** around the ANT SO down states (mean coupling direction: $-175.22^\circ \pm 6.92^\circ$; Rayleigh test, one-sided: $z = 9.33$, $p < 0.0001$). The preferred phases of SO-spindle modulation differed significantly between the ANT and neocortical sites (Watson-Williams test, two-sided: $F = 9.31$; $p = 0.0069$). (g) Directional SO-spindle coupling as obtained by the phase-slope index (PSI; mean \pm SEM) for ANT (blue) and neocortical (gray) SO-spindles. SO-phases significantly predicted spindle amplitudes both in case of ANT (t-test against zero, two-sided: $p = 0.037$) and neocortical SO-spindle events (t-test against zero, two-sided: $p = 0.044$).

Figure 7. MD spindles lock to early SO phases. (a) Occurrence probabilities of **neocortical spindle onsets** with respect to SO down states (bin size = 50 ms), indicating that **spindles started at the SO down-to-up transition** (0.05 to 0.15 sec; dependent-samples t-test, two-sided, $p < 0.0001$; corrected across time; peak: 0.1 sec). (b) **MD spindles started around SO down states** (-0.35 to 0 sec; dependent-samples t-test, two-sided, $p < 0.0001$; corrected across time; peak: -0.05 sec). (c) Time–frequency representation (TFR) of neocortical SO-spindles ($N = 338.5 \pm 79.7$; locked to down states), contrasted against event-free segments. Contour lines indicate clusters (dependent-samples t-test, two-sided: $p < 0.05$, corrected across time and frequency). (d) TFR of all MD SO-spindles (MD: $N = 488 \pm 53.3$; locked to down states), contrasted against event-free segments. Contour lines indicate clusters (dependent-samples t-test, two-sided: $p < 0.05$, corrected across time and frequency). (e) Direct comparison of the occurrence probabilities of MD (red) and neocortical SO-spindles (gray) revealed that MD spindles start at earlier SO phases (-0.35 to 0.05, dependent-samples t-test, two-sided: $p < 0.0001$; corrected across time). (f) Phases of the SO-spindle modulation derived from neocortical (gray) and MD (red) events. **Neocortical spindles started** at the SO down-to-up transition (phase of down states corresponds to $\pm \pi$; mean coupling direction: $-138.43 \pm 9.05^\circ$; Rayleigh test, one-sided: $z = 8.18$, $p < 0.0001$), while **MD spindles started** around the SO down states (mean coupling direction: $172.26 \pm 6.67^\circ$; Rayleigh test, one-sided: $z = 12.10$, $p < 0.0001$). The preferred phases of SO-spindle modulation differed between MD and neocortical sites (Watson-Williams test, two-sided: $F = 17.33$; $p = 0.0003$). (g) Directional SO-spindle coupling obtained by the phase-slope index (PSI; mean \pm SEM) for MD (red) and neocortical (gray) SO-spindles. SO-phases predicted spindle amplitudes in MD (t-test against zero, two-sided: $p = 0.032$) and neocortical SO-spindles (t-test against zero, two-sided: $p = 0.049$). (h + i) Directly comparing ANT- (blue) and MD-derived SO-spindles (red) did not yield a significant difference (h) occurrence probabilities: $p = 0.11$; (i) phase distribution: $F = 1.29$; $p = 0.25$.

Results:

Page 9, line 235:

... To address this, we isolated SO-spindles complexes in ANT as well as scalp recordings and **extracted the onset latencies of spindles relative to their corresponding SO down states** within both regions, respectively.

In line with previous findings^{16,27}, occurrence probabilities of neocortical spindles with respect to neocortical SOs (Fig. 6a), indicated that **spindles started specifically at the SO down-to-up transition** (0.05 to 0.15 s; $p < 0.0001$; peak: 0.1 sec). In contrast, **ANT spindles preferential emerged around ANT SO down states** (-0.15 to 0.25 s; $p = 0.0012$; peak: 0 s; Fig. 6b)...

Page 10, line 280:

... Occurrence probabilities indicated that **neocortical spindles started specifically at the down-to-up transition** of neocortical SOs (0.1 to 0.3 s; $p = 0.003$; peak: 0.1 s; Fig 7a). **MD spindles, however, started preferentially briefly before the MD SO down states** (-0.35 to 0 s; $p < 0.0001$; peak: -0.05 s; Fig 7b)...

Methods:

Page 22, line 684:

...To determine the interplay of SOs and sleep spindles (separately within scalp and thalamic recording), a peri-event histogram of **sleep spindles (onsets)** around SO down states (± 1.5 sec) was created...

Page 23, line 698:

...For the analysis of SO-spindle coupling, we filtered the SO-spindle data (**locked to spindle onsets**) in the SO range (0.3–2 Hz, two-pass Butterworth bandpass filter), applied a Hilbert transform and extracted the instantaneous phase angle. Next, we isolated the SO phase angle **at the time of spindle onsets**...